# Sample Reward Soups: Query-efficient Multi-Reward Guidance for Text-to-Image Diffusion Models

**Yinghua Yao**[12] , **Yuangang Pan**[12*]**, Guoji Fu**[3] **, Ivor W. Tsang**[124]

[1]Centre for Frontier AI Research, Agency for Science, Technology and Research, Singapore
[2]Institute of High Performance Computing, Agency for Science, Technology and Research, Singapore
[3]Department of Computer Science, National University of Singapore, Singapore
[4]College of Computing and Data Science, Nanyang Technological University, Singapore
`{yao_yinghua,pan_yuangang,ivor_tsang}@a-star.edu.sg`
`guoji.fu@u.nus.edu`

## Abstract

Recent advances in inference-time alignment of diffusion models have shown reduced susceptibility to reward over-optimization. However, when aligning with multiple black-box reward functions, the number of required queries grows exponentially with the number of reward functions, making the alignment process highly inefficient. To address the challenge, we propose the first inference-time soup strategy, named Sample Reward Soups (SRSoup), for Pareto-optimal sampling across the entire space of preferences. Specifically, at each denoising step, we independently steer multiple denoising distributions using reward-guided search gradients (one for each reward function) and then linearly interpolate their search gradients. This design is effective because sample rewards can be shared when two denoising distributions are close, particularly during the early stages of the denoising process. As a result, SRSoup significantly reduces the number of queries required in the early stages without sacrificing performance. Extensive experiments demonstrate the effectiveness of SRSoup in aligning T2I models with diverse reward functions, establishing a practical and scalable solution. The code is available at `https://github.com/EvaFlower/Sample-Reward-Soups-ICLR26`.

## 1 Introduction

Text-to-Image (T2I) diffusion models have emerged at the forefront of high-quality image synthesis conditioned on text prompts, pushing forward various applications in fields such as art, design, and beyond (Rombach et al., 2022; Ramesh et al., 2022; Saharia et al., 2022; Podell et al., 2024; Chang et al., 2025). Despite their success, controlling T2I models to meet complex user objectives, such as compressibility and subjective aesthetics, remains challenging through simple text prompting alone. This has driven interest in techniques for fine-tuning T2I diffusion models with reinforcement learning (Black et al., 2024), differentiable reward functions (Prabhudesai et al., 2023; Clark et al., 2024), and direct preference optimization (Wallace et al., 2024; Yang et al., 2024a).

Multiple reward functions are often used to provide supervision for fine-tuning. A common approach is to assign different weights to these functions to account for diverse human preferences regarding different reward criteria. This, however, incurs substantial computational costs due to the large space of possible weight combinations. One line of research (Rame et al., 2023) adopted the idea of model soups (Wortsman et al., 2022) to reduce the fine-tuning cost to scale linearly with the number of reward functions. Another effort (Yang et al., 2024b) is to fine-tune a model that conditions multiple reward values in the prompts using supervised learning. Nevertheless, these fine-tuning strategies may lead to reward over-optimization, where the fine-tuned models lose generation quality or diversity (Gao et al., 2023; Jena et al., 2024). Additionally, these models may not generalize well to unseen prompt distributions that drift far from their training set.

---

*Corresponding author.

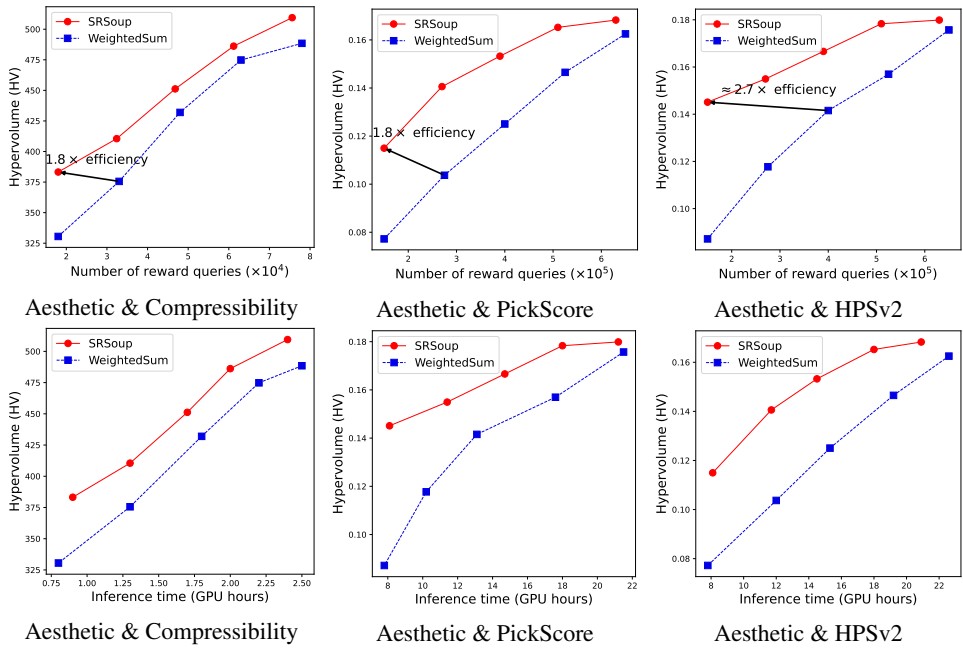

Figure 1: Quantitative evaluation of two-objective optimization, showing that our SRSoup is more query-efficient and achieves comparable results with less reward queries and inference time.

In this work, instead of fine-tuning models for alignment, we explore guiding the denoising process of diffusion models using multi-reward signals. This type of inference-time alignment has been shown to be less prone to reward over-optimization (Kim et al., 2025; Eyring et al., 2024; Yeh et al., 2025). However, a main challenge in the multi-reward setting is that traversing weighted combinations of multiple reward objectives often requires a large number of reward queries. To address this, we propose Sample Reward Soups (SRSoup) as a query-efficient solution. Specifically, we begin by optimizing the stepwise denoising distributions of diffusion models using reward-guided search gradients for each individual reward function. Then, we interpolate these search gradients to approximate the search gradient corresponding to a weighted sum of the reward functions. The key insight is that the denoising distributions—especially in the early stages of the process—overlap when optimized from the same noise point under different weightings of multiple rewards (Fig. 2). We summarize our contributions as follows:

- We advocate a multi-reward alignment approach at inference time of diffusion models to enable Pareto-optimal sampling across the entire preference space w.r.t. different reward functions.
- We propose Sample Reward Soups (SRSoup), which significantly reduces the number of reward queries compared to the weighted sum strategy.
- We empirically demonstrate that our approach obtains better solutions than fine-tuning-based rewarded soup variants and achieves comparable performance to the weighted-sum strategy while requiring fewer reward queries in Fig. 1.

## 2 RELATED WORK

**Fine-tuning-based Alignment.** Fine-tuning T2I diffusion models using reinforcement learning (RL) has been proposed to align them with arbitrary black-box reward functions (Black et al., 2024; Fan et al., 2024; Deng et al., 2024). These approaches interpret the reverse diffusion process as a multi-step decision-making process and apply policy gradient algorithms for fine-tuning based on pre-specified rewards. In parallel, direct back-propagation through differentiable reward functions has been proposed for T2I model alignment (Clark et al., 2024; Prabhudesai et al., 2023; Xu et al., 2024). In addition, reward fine-tuning with supervised learning has been proposed (Lee et al., 2023; Wu et al., 2023; Dong et al., 2023). In the approaches, images are generated with the pre-trained diffusion model, and the model is then fine-tuned using a weighted likelihood, where the weights are determined by the rewards or by a selected set of images with high rewards. Another popular approach involves fine-tuning T2I models directly on preference data using a Direct Preference Optimization (DPO)

objective (Yang et al., 2024a; Wallace et al., 2024; Liang et al., 2024). Venkatraman et al. (2024) casts reward fine-tuning as a stochastic optimal control (SOC) problem and proposing entropy-regularized SOC for better sample quality and diversity. Upon it, Domingo-Enrich et al. (2025) further proposes a more simplified and elegant solution, Adjoint Matching, to solve SOC problems. Recent works (Wang et al., 2025; Liu et al., 2025) apply group relative policy optimization (GRPO) on flow-based T2I models for reward alignment, establishing a promising foundation for this direction.

**Alignment at Inference Time.** Diffusion model exhibits flexibility in its inference, beyond large language models, allowing for the incorporation of more controls during the inference stage. Classifier guidance (Dhariwal & Nichol, 2021) is a popular strategy to steer the denoising process in alignment with the target objective by adding an auxiliary objective gradient to the score function of diffusion models. Many works (Wallace et al., 2023; Bansal et al., 2023; Eyring et al., 2024; Kim et al., 2025) have proposed incorporating guidance from a differentiable reward model at the inference stage of T2I models. A recent work (Yeh et al., 2025) proposed optimizing noise distribution for the diffusion sampling based on rewards from a black-box function. DyMO (Xie & Gong, 2025) focuses on improving semantic alignment in the single-reward setting by introducing a dedicated semantic alignment objective.

**Multi-reward Alignment.** Weighted sum is a popular method for combining multiple reward functions into a single one, which can then be used to fine-tune models (Li et al., 2020; Zhou et al., 2024). However, this approach typically requires using many different sets of weights to obtain Pareto-optimal solutions for the multiple objectives, resulting in significant computational overhead. Rewarded soups (Rame et al., 2023) has been proposed to reduce the cost (Clark et al., 2024; Prabhudesai et al., 2023). It first obtains models fine-tuned independently on each reward function and then interpolates the model weights to find diverse solutions that account for various trade-offs among the objectives. Supervised fine-tuning methods have also been proposed for multi-reward alignment (Lee et al., 2024; Yang et al., 2024b). Specifically, (Lee et al., 2024) generates images with a pre-trained T2I model and applies multiple reward functions to select a Pareto set of images. The T2I model is then fine-tuned using RL based on this Pareto set of samples. Similarly, (Yang et al., 2024b) generates images with the pre-trained model, applies multiple reward functions to score the images, and extends the text inputs with their corresponding reward vectors to fine-tune the T2I model. The Pareto set of images is then obtained by specifying the reward vectors during the inference process. In summary, existing approaches either require fine-tuning multiple models for multi-objective alignment or rely on large supervised datasets. These fine-tuning methods often suffer from issues of reward over-optimization, leading to a loss in generation quality and diversity (Gao et al., 2023; Jena et al., 2024). It is also possible to apply the weighted sum to provide multi-reward guidance at inference time in diffusion models (Kim et al., 2025). However, the need for numerous reward queries across various combinations of weightings for different reward objectives leads to a high computational cost. This work will target a query-efficient multi-reward alignment method without any training of diffusion models.

## 3 PRELIMINARIES

**Diffusion Models** (Ho et al., 2020; Yang et al., 2022) progressively diffuse data to noise, and then learn to reverse this process for sample generation. Considering a sequence of prescribed noise scales $0 < \beta_1 < \beta_2 < \ldots < \beta_T < 1$, clean data $\boldsymbol{x}_0 \sim q_{\text{data}}(\boldsymbol{x})$ are diffused to noise $\boldsymbol{x}_T \sim \mathcal{N}(\boldsymbol{0}, \boldsymbol{I})$ via constructing a Markov chain $\{\boldsymbol{x}_0, \boldsymbol{x}_1, \ldots, \boldsymbol{x}_T\}$, where $q(\boldsymbol{x}_t|\boldsymbol{x}_{t-1}) = \mathcal{N}(\boldsymbol{x}_t; \sqrt{1-\beta_t}\boldsymbol{x}_{t-1}, \beta_t\boldsymbol{I})$. In particular, $q(\boldsymbol{x}_t|\boldsymbol{x}_0) = \mathcal{N}(\boldsymbol{x}_t; \sqrt{\alpha_t}\boldsymbol{x}_0, (1-\alpha_t)\boldsymbol{I})$, where $\alpha_t = \prod_{i=1}^{t}(1-\beta_t)$. The training of the reverse diffusion process is to minimize a simplified variational bound of negative log likelihood. Namely,

$$\mathbb{E}_{\boldsymbol{x}_0 \sim q_{\text{data}}(\boldsymbol{x}), \epsilon \sim \mathcal{N}(\boldsymbol{0}, \boldsymbol{I})} \left[ \|\boldsymbol{\epsilon} - \boldsymbol{\epsilon}_\theta(\boldsymbol{x}_t, t)\|^2 \right], \quad (1)$$

where $\boldsymbol{x}_t = \sqrt{\alpha_t}\boldsymbol{x}_0 + \sqrt{1-\alpha_t}\boldsymbol{\epsilon}$, and $\boldsymbol{\epsilon}_\theta(\boldsymbol{x}_t, t)$ is a neural network-based approximator to predict the noise $\boldsymbol{\epsilon}$ from $\boldsymbol{x}_t$. After obtaining the optimal noise predictor $\boldsymbol{\epsilon}_\theta(\boldsymbol{x}_t, t)$, samples can be generated by starting from $\boldsymbol{x}_T \sim \mathcal{N}(\boldsymbol{0}, \boldsymbol{I})$ and reversing the Markov chain following the step-wise sampling distribution as below:

$$\boldsymbol{x}_{t-1} \sim \mathcal{N}(\boldsymbol{x}_{t-1}; \mu_\theta(\boldsymbol{x}_t, t), \tilde{\beta}_t \boldsymbol{I}),$$
$$\mu_\theta(\boldsymbol{x}_t, t) = \frac{1}{\sqrt{1-\beta_t}}\left(\boldsymbol{x}_t - \frac{\beta_t}{\sqrt{1-\alpha_t}}\boldsymbol{\epsilon}_\theta(\boldsymbol{x}_t, t)\right), \quad (2)$$

---

**Algorithm 1** Reward-guided Search Gradient for Denoising Distribution Optimization

---
1: **Input:** reward function $f(\boldsymbol{x})$, denoising distribution $\mathcal{N}(\boldsymbol{x}_{t-1}; \boldsymbol{\mu}_{t-1}, \beta_t \boldsymbol{I})$, random noises $\{\boldsymbol{z}^n\}_{n=1}^N$ from $\mathcal{N}(\boldsymbol{0}, \boldsymbol{I})$, step size $\tau_t$.
2: Construct $N$ denoised samples $\{\boldsymbol{x}_{t-1}^n | \boldsymbol{x}_{t-1}^n = \boldsymbol{\mu}_{t-1} + \sqrt{\beta_t} \boldsymbol{z}^n\}_{n=1}^N$.
3: Obtain the black-box rewards $\{f(\boldsymbol{x}_{t-1}^n)\}_{n=1}^N$.
4: Estimate the gradient $\nabla_{\boldsymbol{\mu}_{t-1}} \mathcal{F}(\boldsymbol{\mu}_{t-1}) = \frac{1}{\sqrt{\beta_t} N} \sum_{n=1}^N f(\boldsymbol{x}_{t-1}^n) \boldsymbol{z}^n$.
5: Obtain the high-reward mean $\bar{\boldsymbol{\mu}}_{t-1} = \boldsymbol{\mu}_{t-1} + \tau_t \nabla_{\boldsymbol{\mu}_{t-1}} \mathcal{F}(\boldsymbol{\mu}_{t-1})$.
6: **Output:** search gradient $\nabla_{\boldsymbol{\mu}_{t-1}} \mathcal{F}(\boldsymbol{\mu}_{t-1})$, optimal mean $\bar{\boldsymbol{\mu}}_{t-1}$ and rewards $\{f(\boldsymbol{x}_{t-1}^n)\}_{n=1}^N$.

---

where $t = T, T-1, \ldots, 1$. The variance $\tilde{\beta}_t$ can be set to either $\beta_t$ or $\frac{1-\alpha_{t-1}}{1-\alpha_t} \beta_t$ (Ho et al., 2020). In the following, we set $\tilde{\beta}_t$ as $\beta_t$.

**Multi-objective Optimization** Let $\boldsymbol{x} \in \mathbb{R}^D$ be a sample of interest, e.g., an animal of specific appearances, or a material with desired functions. Assuming that $F(\boldsymbol{x}) = [f_1(\boldsymbol{x}), f_2(\boldsymbol{x}), \ldots, f_M(\boldsymbol{x})]$ be a set of $M$ rewards, each of which represents a property and is preferred to have a smaller value. The multi-objective optimization problem (Chinchuluun & Pardalos, 2007; Yao et al., 2024) can be conventionally expressed as:

$$\min_{\boldsymbol{x} \in \mathbb{R}^D} F(\boldsymbol{x}) = \min_{\boldsymbol{x} \in \mathbb{R}^D} (f_1(\boldsymbol{x}), f_2(\boldsymbol{x}), \ldots, f_M(\boldsymbol{x})). \tag{3}$$

In the context of multi-objective optimization, for any $\boldsymbol{x}_1, \boldsymbol{x}_2 \in \mathbb{R}^D$, $\boldsymbol{x}_1$ is said to dominate $\boldsymbol{x}_2$, i.e., $\boldsymbol{x}_1 \prec \boldsymbol{x}_2$, iff $f_i(\boldsymbol{x}_1) \le f_i(\boldsymbol{x}_2), \forall i = 1, 2, \ldots, M$, and $F(\boldsymbol{x}_1) \ne F(\boldsymbol{x}_2)$. A point $\boldsymbol{x}^*$ is called Pareto optimal iff there is no any other $\boldsymbol{x}' \in \mathbb{R}^D$ such that $\boldsymbol{x}' \prec \boldsymbol{x}^*$.

## 4 SAMPLE REWARD SOUPS

In this section, we first propose a training-free guidance strategy for multi-objective alignment of diffusion models. Building on it, we further propose *sample reward soups* to improve the query-efficiency of multi-objective guidance.

### 4.1 REWARD-GUIDED SAMPLING FOR BLACK-BOX REWARD ALIGNMENT

While fine-tuning T2I diffusion models using RL for non-differentiable reward optimization has shown success, we in parallel propose a training-free strategy that simply steers the denoising sampling process of diffusion models based on sample rewards.

Specifically, we propose optimizing the denoising distribution $\mathcal{N}(\boldsymbol{x}_{t-1}; \mu_\theta(\boldsymbol{x}_t, t), \beta_t \boldsymbol{I})$ in Equation 2 at each step to encourage high-reward generations. Namely,

$$\max_{\mathcal{N}(\boldsymbol{x}_{t-1}; \mu_\theta(\boldsymbol{x}_t, t), \beta_t \boldsymbol{I})} \mathbb{E}[f(\boldsymbol{x}_{t-1})], \quad \forall t = T, T-1, \ldots, 1. \tag{4}$$

This idea is inherited from black-box optimization Wierstra et al. (2014b); Nesterov & Spokoiny (2017), which enables reward-guided optimization without requiring gradients from the reward function. For simplicity, we only optimize the mean of the Gaussian distribution. Defining $\boldsymbol{\mu}_{t-1} = \mu_\theta(\boldsymbol{x}_t, t)$, we have the following objective:

$$\max_{\boldsymbol{\mu}_{t-1}} \mathcal{F}(\boldsymbol{\mu}_{t-1}) = \mathbb{E}_{\mathcal{N}(\boldsymbol{x}_{t-1}; \boldsymbol{\mu}_{t-1}, \beta_t \boldsymbol{I})}[f(\boldsymbol{x}_{t-1})], \quad \forall t = T, T-1, \ldots, 1. \tag{5}$$

**Theorem 1** (Reward-guided search gradient). *The gradient of $\mathcal{F}(\boldsymbol{\mu}_{t-1})$ w.r.t. a learnable $\boldsymbol{\mu}_{t-1}$ can be obtained as follows:*

$$\nabla_{\boldsymbol{\mu}_{t-1}} \mathcal{F}(\boldsymbol{\mu}_{t-1}) = \frac{1}{\sqrt{\beta_t}} \mathbb{E}_{\mathcal{N}(\boldsymbol{x}; \boldsymbol{\mu}_{t-1}, \beta_t \boldsymbol{I}_d)} \left[ \nabla_{\boldsymbol{x}} f(\boldsymbol{x}) \right] = \frac{1}{\sqrt{\beta_t}} \mathbb{E}_{\mathcal{N}(\boldsymbol{z}; \boldsymbol{0}, \boldsymbol{I})} \left[ f(\boldsymbol{\mu}_{t-1} + \sqrt{\beta_t} \boldsymbol{z}) \boldsymbol{z} \right]. \tag{6}$$

See Appendix A.1 for a detailed derivation.

Equation 6 provides a solution to determine the optimal mean parameter $\bar{\mu}_{t-1}$ in Equation 5 using sample rewards. To avoid computational overhead, we adopt a single-step gradient descent. Namely,

$$\bar{\mu}_{t-1} = \frac{1}{\sqrt{1-\beta_t}} \left( \bar{x}_t - \frac{\beta_t}{\sqrt{1-\alpha_t}} \epsilon_\theta(\bar{x}_t, t) \right) + \tau_t \cdot \nabla_{\mu_{t-1}} \mathcal{F}(\mu_{t-1}), \qquad (7)$$

where $\tau_t$ is the step size and is set to $\beta_t$ for compatible with the original sampling schedule. With a slightly abused notation, we let $\mu_{t-1} = \mu_\theta(\bar{x}_t, t)$, then $\mu_{t-1} = \frac{1}{\sqrt{1-\beta_t}} \left( \bar{x}_t - \frac{\beta_t}{\sqrt{1-\alpha_t}} \epsilon_\theta(\bar{x}_t, t) \right)$ and $\bar{\mu}_t = \mu_{t-1} + \tau_t \cdot \nabla_{\mu_{t-1}} \mathcal{F}(\mu_{t-1})$. Finally, a sample with a high reward can be sampled via: $\bar{x}_{t-1} \sim \mathcal{N}(\bar{\mu}_{t-1}, \beta_t \boldsymbol{I})$.

In practice, we use Monte Carlo approximation for Equation 6. Specifically, at each denoising step $t$, we first draw $N$ standard Gaussian noise $z^n \sim \mathcal{N}(\boldsymbol{0}, \boldsymbol{I})$ and obtain the denoise sample $x_{t-1}^n$ via the transformation $x_{t-1}^n = \mu_{t-1} + \sqrt{\beta_t} z^n$. Then, we obtain the rewards $\{f(x_{t-1}^n)\}_{n=1}^N$, which will be used to update $\mu_{t-1}$ towards $\bar{\mu}_{t-1}$ via Equation 7. To avoid extra noise distraction (Prabhudesai et al., 2023; Clark et al., 2024), we adopt the deterministic formulation of the reverse generative process, i.e., Denoising Diffusion Implicit Models (DDIM) Song et al. (2020) to obtain $\bar{x}_{t-1}$, namely, $\bar{x}_{t-1} = \bar{\mu}_{t-1}$. The whole process is summarized in Algorithm 1.

## 4.2 WEIGHTED SUM FOR MULTI-REWARD ALIGNMENT

Weighted sum is a commonly used strategy to solve the multi-objective optimization. Let $w_{1:M}$ denote $\{w_i\}_{i=1}^M$ for short. Each point $w_{1:M} \in \Delta^{M-1}$ lies on the standard $(M-1)$-simplex: $w_i \geq 0$, $\sum_{i=1}^M w_i = 1$. Given a weight vector $w_{1:M}$, the multi-objective problem (Equation 3) is formulated into a weighted objective as follows:

$$F(\boldsymbol{x}, w_{1:M}) = \sum_{i=1}^M w_i f_i(\boldsymbol{x}) = w_1 f_1(\boldsymbol{x}) + w_2 f_2(\boldsymbol{x}) + \cdots + w_M f_M(\boldsymbol{x}), \qquad (8)$$

where a larger $w_i$ places more emphasis on the corresponding reward function $f_i$.

Following the reward-guided sampling introduced in Section 4.1, we can apply weighted rewards to obtain samples for multi-reward alignment by replacing $f$ with $F$. Namely,

$$\max_{\mu_{t-1}} \mathcal{F}(\mu_{t-1}, w_{1:M}) = \mathbb{E}_{\mathcal{N}(x_{t-1}; \mu_{t-1}, \beta_t \boldsymbol{I})}[F(\boldsymbol{x}_{t-1}, w_{1:M})], \quad \forall t = T, T-1, \ldots, 1. \qquad (9)$$

To accommodate different trade-offs among objectives, we need to construct a set of $L(\gg M)$ weight vectors $\{w_{1:M}^j\}_{j=1}^L$. These weights yield $L$ samples $\{x_0^j\}_{j=1}^L$, which collectively approximate the Pareto front cross multiple objectives. However, directly evaluating all $L$ weighted combinations requires $NTM(L-M+1)$ reward evaluations per prompt. This is because each $w_{1:M}^j$ typically has support over $M$ objectives. Among these, the $M$ one-hot weight vectors each require only $NT$ evaluations, while the remaining $L-M$ combinations require $NTM$ evaluations per prompt. Although this approach mitigates the high training cost of diffusion models, the large number of required reward evaluations can significantly increase inference-time sampling cost, especially when the black-box reward is expensive to compute.

## 4.3 SAMPLE REWARD SOUPS FOR MULTI-REWARD ALIGNMENT

Inspired by Rewarded Soups (Rame et al., 2023), which interpolates the weights of diffusion models fine-tuned on distinct reward functions to approximate a model optimized for their weighted sum, we propose Sample Reward Soups (SRSoup) in the training-free scenario. Our SRSoup is a sample-level analogue that interpolates reward-guided search gradients (Equation 6), derived from distinct reward functions at each denoising step, to approximate the search gradient corresponding to their weighted combination.

**Definition 2.** *Let $\{c_{t-1}^m\}_{m=1}^M$ denote a set of exemplars, each individually guided by a distinct reward function. These exemplars serve as the mean parameters in the denoising distribution, similar to the $\mu_{t-1}$ in Eq. 4. As only one reward function is active, the weight vector of each exemplar becomes one-hot. Therefore, for each exemplar $c_{t-1}^m$, Eq. 9 can be simplified as:*

$$\max_{c_{t-1}^m} \mathcal{F}(c_{t-1}^m, e_m) = \mathbb{E}_{\mathcal{N}(x_{t-1}; c_{t-1}^m, \beta_t \boldsymbol{I})}[f_m(\boldsymbol{x}_{t-1})], \quad \forall t = T, T-1, \ldots, 1, \qquad (10)$$

---

**Algorithm 2** Sample Reward Soups (SRSoup) at the Denoising Step $t$

---

1: **Input:** $M$ reward functions $\{f_1, f_2, \ldots, f_M\}$, $L$ weights $\{w_{1:M}^l\}_{l=1}^L$, #query $N$, denoising timestep $t$, step size $\tau_t$.
2: **if** $t == T$ **then**
3:     Initialize $L$ samples $\{\bar{\boldsymbol{x}}_T^l\}_{l=1}^L$ using a same point from $\mathcal{N}(\boldsymbol{0}, \boldsymbol{I})$.
4: **else**
5:     $\{\bar{\boldsymbol{x}}_t^l\}_{l=1}^L$ obtained by using SRSoup at the denoising step $t+1$ (Algorithm 2).
6: **end if**
7: Sample $N$ random noises $\{\boldsymbol{z}^n\}_{n=1}^N$ from $\mathcal{N}(\boldsymbol{0}, \boldsymbol{I})$.
   ### Obtain search gradients w.r.t. each reward function based on the top $M$ samples $\{\bar{\boldsymbol{x}}_t^l\}_{l=1}^M$, with one-hot weights $e_m$ (1 at the $m$-th entry).
8: **for** $m = 1, 2, \ldots, M$ **do**
9:     Obtain the denoised mean (exemplar in Definition 2): $\boldsymbol{c}_{t-1}^m = \frac{1}{\sqrt{1-\beta_t}}\left(\bar{\boldsymbol{x}}_t^m - \frac{\beta_t}{\sqrt{1-\alpha_t}}\boldsymbol{\epsilon}_\theta(\bar{\boldsymbol{x}}_t^m, t)\right)$.
10:    Obtain the search gradient $\nabla_{\boldsymbol{c}_{t-1}^m}\mathcal{F}(\boldsymbol{c}_{t-1}^m, e_m)$, the optimal mean $\bar{\boldsymbol{c}}_{t-1}^m$ and the black-box rewards $\{f_m(\boldsymbol{x}_{t-1}^{m,n})\}_{n=1}^N$ for $\mathcal{N}(\boldsymbol{x}_{t-1}^m; \boldsymbol{c}_{t-1}^m, \beta_t\boldsymbol{I})$ with $f_m$ and $\{\boldsymbol{z}^n\}_{n=1}^N$ using Algorithm 1.
11:    Obtain the high-reward denoised sample: $\bar{\boldsymbol{x}}_{t-1}^m = \bar{\boldsymbol{c}}_{t-1}^m$.
12: **end for**
    ### Interpolate search gradients.
13: **for** $l = M+1, M+2, \ldots, L$ **do**
14:    Obtain the denoised mean: $\boldsymbol{\mu}_{t-1}^l = \frac{1}{\sqrt{1-\beta_t}}\left(\bar{\boldsymbol{x}}_t^l - \frac{\beta_t}{\sqrt{1-\alpha_t}}\boldsymbol{\epsilon}_\theta(\bar{\boldsymbol{x}}_t^l, t)\right)$.
15:    Calculate the interpolated gradient:

$$\nabla_{\boldsymbol{\mu}_{t-1}^l}\mathcal{F}(\boldsymbol{\mu}_{t-1}^l, w_{1:M}^l) = \sum_{m=1}^M w_m^l\left(\nabla_{\boldsymbol{c}_{t-1}^m}\mathcal{F}(\boldsymbol{c}_{t-1}^m, e_m) + \frac{1}{\sqrt{\beta_t}N}\sum_{n=1}^N f_m(\boldsymbol{x}_{t-1}^{m,n})(\boldsymbol{c}_{t-1}^m - \boldsymbol{\mu}_{t-1}^l)\right)$$

16:    Obtain the high-reward denoised sample: $\bar{\boldsymbol{x}}_{t-1}^l = \bar{\boldsymbol{\mu}}_{t-1}^l = \boldsymbol{\mu}_{t-1}^l + \tau_t\nabla_{\boldsymbol{\mu}_{t-1}^l}\mathcal{F}\left(\boldsymbol{\mu}_{t-1}^l, w_{1:M}^l\right)$.
17: **end for**
18: **Output:** $\{\bar{\boldsymbol{x}}_{t-1}^l\}_{l=1}^L$ that optimizes $M$ reward functions with respective weights $\{w_{1:M}^l\}_{l=1}^L$.

---

*where $m = 1, 2, \ldots, M$ and $e_m \in \Delta^{M-1}$ is a one-hot vector with the $m$-th entry equal to 1.*     □

**Proposition 3.** *Given a set of exemplars $\{\boldsymbol{c}_{t-1}^m\}_{m=1}^M$ introduced in definition 2. Let $\delta_m = \boldsymbol{\mu}_{t-1} - \boldsymbol{c}_{t-1}^m$ $\forall m = 1, 2, \ldots, M$, the gradient of $\boldsymbol{\mu}_{t-1}$ in Equation 9 can be approximated by the gradients of these exemplars as follows:*

$$\nabla_{\boldsymbol{\mu}_{t-1}}\mathcal{F}(\boldsymbol{\mu}_{t-1}, w_{1:M}) = \sum_{m=1}^M w_m\left(\nabla_{\boldsymbol{c}_{t-1}^m}\mathcal{F}(\boldsymbol{c}_{t-1}^m, e_m) + \nabla_{\boldsymbol{c}_{t-1}^m}^2\mathcal{F}(\boldsymbol{c}_{t-1}^m, e_m)\delta_m + \mathcal{O}(\|\delta_m\|^2)\right),$$

*See Appendix A.2 for a detailed derivation.*     □

According to proposition 3, the gradient approximation by $\nabla_{\boldsymbol{c}_{t-1}^m}\mathcal{F}(\boldsymbol{c}_{t-1}^m, e_m)$ is accurate only when the difference $\delta_m$ is very small. Otherwise, we need to calculate the second derivative $\nabla_{\boldsymbol{c}_{t-1}^m}^2\mathcal{F}(\boldsymbol{c}_{t-1}^m, e_m)$, which is however computationally expensive. In the following, we derive another correction term to reduce the approximation gap without calculating the second derivative.

**Proposition 4.** *Given two Gaussian distributions $P = \mathcal{N}(\boldsymbol{c}_{t-1}^m, \beta_t\boldsymbol{I})$ and $Q = \mathcal{N}(\boldsymbol{\mu}_{t-1}, \beta_t\boldsymbol{I})$. Suppose the Euclidean distance between the means is bounded as $\|\boldsymbol{c}_{t-1}^m - \boldsymbol{\mu}_{t-1}\| \leq \varepsilon$. Then, the total variation distance (TV) between the product distributions of $N$ independent samples satisfies $\mathrm{TV}(P^{\otimes N}, Q^{\otimes N}) \leq \frac{N\varepsilon}{\sqrt{4\beta_t}}$. See Appendix A.3 for a detailed derivation.*     □

Proposition 4 relaxes the assumption of closeness between two points to the overlap between two distributions. It shows that samples drawn from $\mathcal{N}(\boldsymbol{c}_{t-1}^m, \beta_t\boldsymbol{I})$ can be statistically indistinguishable from those drawn from $\mathcal{N}(\boldsymbol{\mu}_{t-1}, \beta_t\boldsymbol{I})$ under certain conditions. Motivated by this, our Sample Reward Soup method reuses the search gradients computed solely from exemplars to achieve multi-reward alignment, leveraging the distributional overlap to ensure valid gradient estimation.

**Sample Reward Soup (SRSoup):** Given a set of exemplars $\{\boldsymbol{c}_{t-1}^m\}_{m=1}^M$ introduced in definition 2, let $\{\boldsymbol{x}_{t-1}^{m,n}\}_{n=1}^N$ denote a set of samples drawn from the exemplar distribution $\mathcal{N}(\boldsymbol{c}_{t-1}^m, \beta_t\boldsymbol{I})$

and $\{f_m(\boldsymbol{x}_{t-1}^{m,n})\}_{n=1}^N$ be the corresponding rewards. Following proposition 4, each component $\nabla_{\boldsymbol{\mu}_{t-1}}\mathcal{F}(\boldsymbol{\mu}_{t-1}, e_m)$ in Equation 3 can be further approximated as

$$\nabla_{\boldsymbol{\mu}_{t-1}}\mathcal{F}(\boldsymbol{\mu}_{t-1}, e_m) \approx \nabla_{\boldsymbol{c}_{t-1}^m}\mathcal{F}\left(\boldsymbol{c}_{t-1}^m, e_m\right)$$
$$+ \frac{1}{\sqrt{\beta_t}N}\sum_{n=1}^N f(\boldsymbol{x}_{t-1}^{m,n})(\boldsymbol{c}_{t-1}^m - \boldsymbol{\mu}_{t-1}), \tag{11}$$

where the second term is regarded as the correction term for the gradient approximation.

Furthermore, we propose two strategies to enforce overlap between the distributions. The first is to initialize from a same noise sample, $\boldsymbol{x}_T \sim \mathcal{N}(\boldsymbol{0}, \boldsymbol{I})$, so that $\boldsymbol{\mu}_{T-1} = \boldsymbol{c}_{T-1}^1 = \ldots = \boldsymbol{c}_{T-1}^M$ (i.e., the initialization stage in Fig. 2). The second is to sample the same set of noises to construct inputs for querying the $M$ reward functions. This minimizes divergence among exemplars optimized for different reward functions, thereby reducing the gap between $\boldsymbol{\mu}_{t-1}$ and $\{\boldsymbol{c}_{t-1}^m\}_{m=1}^M$.

Nevertheless, as $\beta_t$ gradually decreases over time in the denoising process, the distributions may no longer overlap. So we adopt a hybrid strategy. At the early stage of the denoising process (i.e., the red Gaussians in Fig. 2), where the distribution $\mathcal{N}(\boldsymbol{\mu}_{t-1}, \beta_t\boldsymbol{I})$ exhibits overlap with the exemplar distributions $\mathcal{N}(\boldsymbol{c}_{t-1}^m, \beta_t\boldsymbol{I})$ for all $m = 1, 2, \ldots, M$, we perform Sample Reward Soups (Algorithm 2). At the later stage of the denoising process (i.e., the green Gaussians in Fig. 2), where the distribution $\mathcal{N}(\boldsymbol{\mu}_{t-1}, \beta_t\boldsymbol{I})$ becomes distinct from the exemplar distributions, we adopt the true weighted reward update w.r.t. Equation 9. Let $K$ denote the number of soup steps (Equation 11). We perform SRSoup sampling for the first $K$ steps (i.e., when $t > T - K$) and use weighted-sum sampling thereafter (i.e., when $t \leq T - K$).

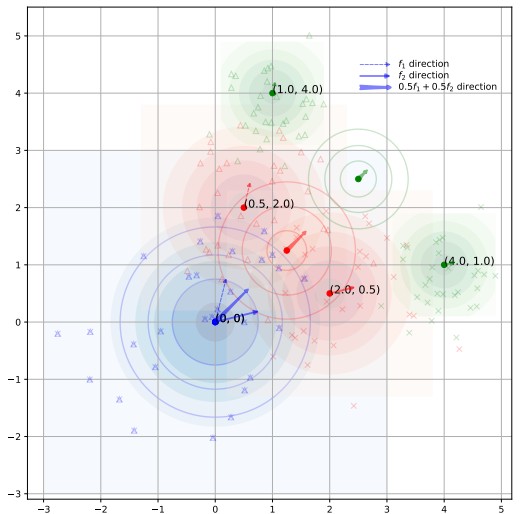

Figure 2: Three blue Gaussians (fully overlapped) denote the initialization stage ($t = T$); three red Gaussians (partially overlapped) denote the early stage of the denoising process; three green Gaussians (non-overlapped) denote the later stage.

**Remark 1.** *In the context of diffusion models, the denoising distributions are isotropic. Therefore, each dimension of $\boldsymbol{\mu}_{t-1}$ can be treated as updated independently. Under this setting, our assumption about the overlap of the denoising distributions can be considered dimension-wise, making it easier to hold, as it avoids the curse of dimensionality. Empirically, we adopt the Bhattacharyya coefficient (Bhattacharyya, 1943) to evaluate the average dimension-wise difference between the distributions at each time step, which aligns with our theoretical analysis. This observation further supports the feasibility and validity of our proposed method. Please refer to the Appendix for details.*

## 5 EXPERIMENTS

In this section, we apply our SRSoup to align T2I diffusion models with a diverse variety of reward objectives. Due to space limits, we put more results and additional experiment details in Appendix.

### 5.1 EXPERIMENTAL SETUP

**Datasets.** We conduct experiments on animal prompts from ImageNet (Deng et al., 2009) and prompts from Human Preference Dataset (HPD) (Wu et al., 2023). For fine-tuning methods, we use 45 animal prompts and 750 HPD prompts as the training set following previous settings (Black et al., 2024; Prabhudesai et al., 2023; Zhang et al., 2024). Evaluation is performed on unseen prompts from the datasets.

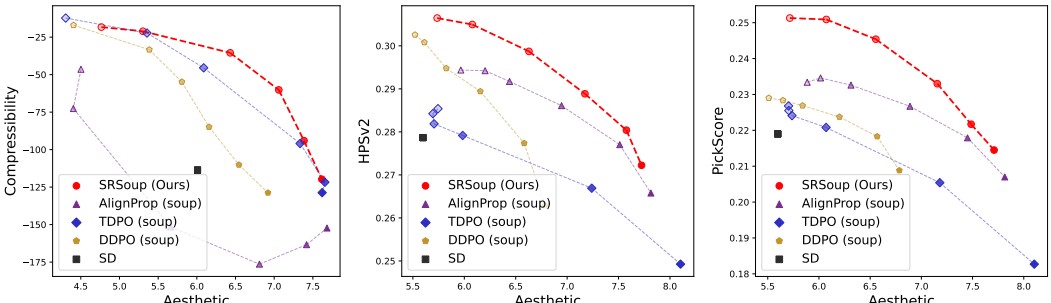

Figure 3: Pareto front comparison of SRSoup and model soup baselines, aligned with two rewards $w_1 f_1 + w_2 f_2$. The depth of the marker color denotes the value of $w_1$; deeper indicates larger.

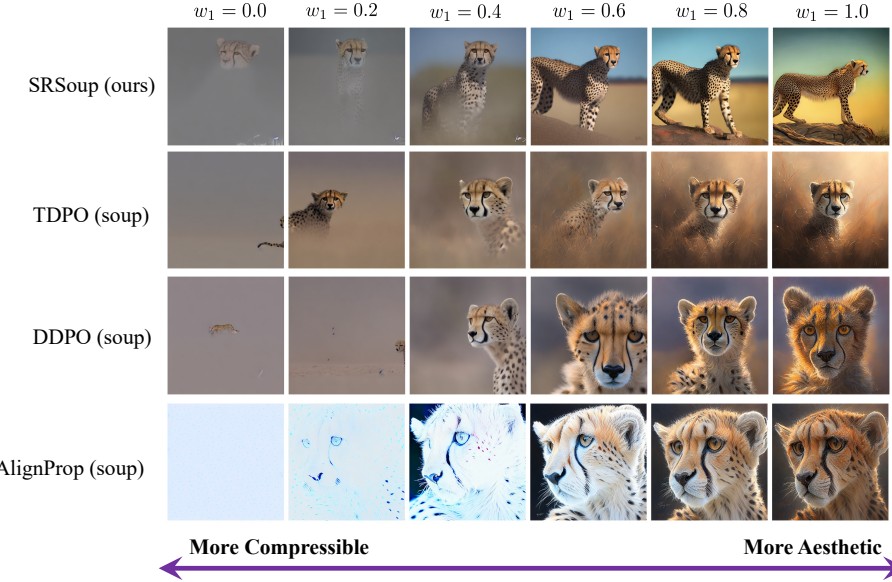

Figure 4: Qualitative comparison of T2I model aligned with the aesthetic reward $f_1$ and the compressibility reward $f_2$. Here, $w_1 * f_1 + (1 - w_1) * f_2$. Prompt: "cheetah".

**Reward Models.** We consider commonly used reward models, namely, compressibility function (Black et al., 2024), LAION aesthetic predictor (Schuhmann et al., 2022), HPSv2 (Wu et al., 2023) and PickScore (Kirstain et al., 2023).

**Baselines.** To demonstrate the superiority of our method, we compare with DDPO (soup) Black et al. (2024), TDPO (soup) (Zhang et al., 2024), and AlignProp (soup) (Prabhudesai et al., 2023). In particular, DDPO and TDPO fine-tune T2I models with RL, while AlignProp fine-tunes models using differentiable reward functions. The soup version means that we first fine-tune models on each reward function independently, obtaining $\{\boldsymbol{\theta}_i\}_{i=1}^{M}$ and then interpolate models by $\{\sum_{i=1}^{M} w_i^l \boldsymbol{\theta}_i\}_{l=1}^{L}$. See Appendix for comparison with other multi-reward fine-tuning methods.

**Hyperparameters.** Unless otherwise specified, we set the number of denoising timesteps $T$ to 50, the number of query samples per reward function at each timestep $N$ to 30, and the number of soup steps $K$ to 20 (introduced in Section 4.3).

## 5.2 COMPARED WITH BASELINES

We adopt Stable Diffusion (SD1.5) (Rombach et al., 2022) as the pre-trained T2I model for all methods. Fig. 3 presents a comparison for two-objective alignment between our SRSoup and various model soup baselines in T2I generation across three scenarios. SRSoup consistently achieves superior Pareto fronts, demonstrating its effectiveness in leveraging reward-guided sampling for multi-objective optimization.

Fig. 4 visualizes the generated images for the prompt "cheetah" under different trade-offs between compressibility and aesthetic quality. Results of other animal prompts are put in Appendix. The fine-tuning methods suffer from over-optimization—a common issue in such approaches (Kim et al., 2025)—which causes the generated images to lose diversity. The images of different prompts optimized for aesthetic reward by DDPO, TDPO and AlignProp show similar backgrounds as shown in Appendix.

**Scaling to three objectives.** To access the scalability of our SRSoup, we optimize three reward functions, i.e., aesthetic score, HPSv2 and PickScore, on HPD prompts. Fig. 5 shows the superiority of our SRSoup on this more challenging scenario. In Fig. 14 (Appendix), our SRSoup achieves the best prompt alignment. TDPO suffers from severe over-optimization, generating images that do not correspond to its prompt, when optimizing aesthetic rewards.

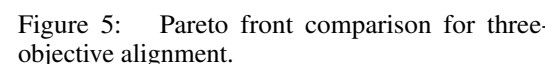
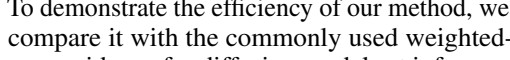

Figure 5: Pareto front comparison for three-objective alignment.

### 5.3 Compared with Weighted sum Guidance under Limited Query Budget

To demonstrate the efficiency of our method, we compare it with the commonly used weighted-sum guidance for diffusion models at inference time, under comparable query budgets. The budget setup is detailed in Appendix. As shown in Fig. 1, our SRSoup consistently achieves better multi-reward optimization while requiring reward queries comparable to weighted-sum guidance. Moreover, under a limited query budget, our SRSoup can achieve comparable performance with $1.8\times$ fewer reward queries for the first two cases and about $2.7\times$ for the last case.

### 5.4 Applied in other T2I Diffusion Models

To demonstrate the generality of our method, we apply our method to SDXL (Podell et al., 2024), which is regarded as the pretrained T2I model. Fig. 6 shows the improvement in image quality with a more advanced Stable Diffusion model. The quantitative results can be seen in Appendix.

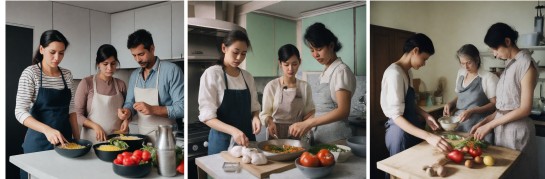

Figure 6: SRSoup with SDXL base for the aesthetic reward function $f_1$ and HPSv2 reward function $f_2$. The weights are $0.0, 0.5, 1.0$, respectively.

### 5.5 Ablation Study

In this section, we study the effects of two hyperparameters in the SRSoup framework: (1) $K$, the number of our soup steps before the true search gradient from the weighted sum is applied; and (2) $N$, the number of queries per reward function at each timestep. We also conduct an ablation on the sample reward soup component.

The left panel of Fig. 7 shows the impact of varying $K$ with $N$ fixed at 30. Even without weighted sum guidance (i.e., $K = 50$), SRSoup achieves a reasonable trade-off between objectives benefiting from the effective interpolated gradients at the early stage. Reducing $K$—incorporating search gradients from the weighted sum earlier in the process—leads to improved performance by leveraging more true rewards. Notably, $K \leq 20$ reduces reward queries substantially (by up to $40\%$) without significantly sacrificing performance. The middle panel varies $N$ with $K$ fixed at 30, demonstrating performance gains with more queries. The right panel highlights the effectiveness of the sample reward soup strategy, which provides informative guidance and yields a significantly better Pareto front than the unconditional sampling counterpart.

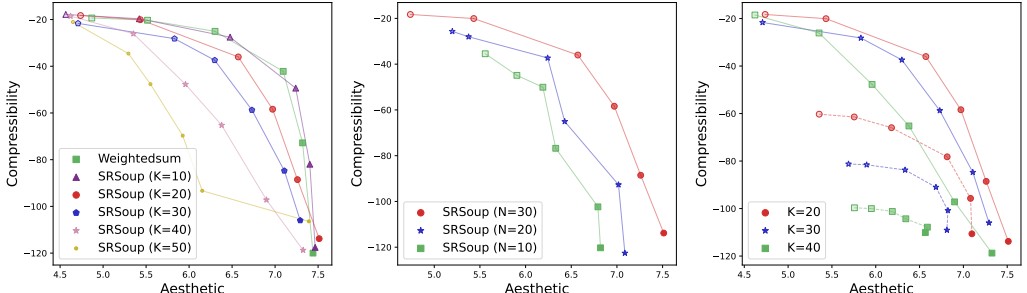

Figure 7: Ablation study of SRSoup. $K$ refers to the number of the soup steps before switching to weighted-sum sampling. $N$ refers to the number of queries per reward per time step. The solid line corresponds to SRSoup with different $K$, while the dashed line represents the method with unconditional sampling for the first $K$ steps and weighted sum sampling thereafter.

## 6  CONCLUSION

We introduce SRSoup, the first training-free soup strategy that interpolates reward-guided search gradients from individual rewards for multiple-reward alignment. We provide a theoretical analysis supporting the effectiveness of SRSoup, which uses a sample reward sharing mechanism fundamentally different from the parameter sharing design of existing model soup approaches. In particular, SRSoup consistently outperforms various model soup baselines in both two-reward and three-reward alignment settings, without suffering from over-optimization. Moreover, it achieves performance comparable to inference-time weighted-sum alignment while requiring fewer reward queries.

## ETHICS STATEMENT

The techniques studied in this paper (multi-reward alignment), while capable of balancing complex and conflicting human values to enable more nuanced, personalized, and safer AI behavior, may also introduce risks such as biased reward weighting, value conflicts, and potential misuse. Ensuring careful design and governance is crucial to maximize societal benefits while minimizing unintended harms.

## REPRODUCIBILITY STATEMENT

Section 5 and Appendix D provide detailed implementation information for our method and baselines, covering hyperparameter settings as well as training and sampling procedures. All experiments use publicly available datasets and evaluation metrics.

## ACKNOWLEDGMENT

This work was supported by the National Research Foundation, Singapore under its National Large Language Models Funding Initiative (AISG Award No: AISG-NMLP-2024-004). Any opinions, findings and conclusions or recommendations expressed in this material are those of the author(s) and do not reflect the views of National Research Foundation, Singapore.

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

## A    DETAILED PROOFS

### A.1    PROOF OF THEOREM 1

*Proof.* The core idea of the proof is to apply the log-derivative trick to decouple the gradient computation from the reward function. To be specific,

$$
\nabla_{\boldsymbol{\mu}_{t-1}} \mathcal{F}(\boldsymbol{\mu}_{t-1}) \overset{①}{=} \mathbb{E}_{\mathcal{N}(\boldsymbol{x}_{t-1};\boldsymbol{\mu}_{t-1},\beta_t \boldsymbol{I}_d)}\Big[ f(\boldsymbol{x}_{t-1})\nabla_{\boldsymbol{\mu}_{t-1}} \log \mathcal{N}(\boldsymbol{x}_{t-1};\boldsymbol{\mu}_{t-1},\beta_t \boldsymbol{I}_d)\Big]
$$

$$
= \frac{1}{\beta_t}\mathbb{E}_{\mathcal{N}(\boldsymbol{x}_{t-1};\boldsymbol{\mu}_{t-1},\beta_t \boldsymbol{I}_d)}\Big[ f(\boldsymbol{x}_{t-1})(\boldsymbol{x}_{t-1} - \boldsymbol{\mu}_{t-1})\Big]
$$

$$
= \frac{1}{\sqrt{\beta_t}}\mathbb{E}_{\mathcal{N}(\boldsymbol{x}_{t-1};\boldsymbol{\mu}_{t-1},\beta_t \boldsymbol{I}_d)}\Big[ f(\boldsymbol{x}_{t-1})\boldsymbol{z}\Big]
$$

where ① applies the log-derivative trick (Wierstra et al., 2014a), which enables gradient estimation with a non-differentiable reward function.

$\square$

### A.2    PROOF OF PROPOSITION 3

**Proposition 3**    Given a set of exemplars $\{c_{t-1}^m\}_{m=1}^M$ introduced in Definition 2. Let $\delta_m = \boldsymbol{\mu}_{t-1} - c_{t-1}^m \ \forall m = 1, 2, \ldots, M$, the gradient of $\boldsymbol{\mu}_{t-1}$ in Equation 9 can be approximated by the gradients of these exemplars as follows:

$$
\nabla_{\boldsymbol{\mu}_{t-1}} \mathcal{F}(\boldsymbol{\mu}_{t-1}, w_{1:M}) = \sum_{m=1}^M w_m \nabla_{\boldsymbol{\mu}_{t-1}} \mathcal{F}(\boldsymbol{\mu}_{t-1}, e_m)
$$

$$
= \sum_{m=1}^M w_m \left( \nabla_{c_{t-1}^m} \mathcal{F}(c_{t-1}^m, e_m) + \nabla_{c_{t-1}^m}^2 \mathcal{F}(c_{t-1}^m, e_m)\delta_m + \mathcal{O}(\|\delta_m\|^2) \right),
$$

*Proof.* Our goal is to approximate the search gradient, guided by weighted rewards, using search gradients computed at each exemplar $c_{t-1}^m$. First, we have

$$
\nabla_{\boldsymbol{\mu}_{t-1}} \mathcal{F}(\boldsymbol{\mu}_{t-1}, w_{1:M}) = \sum_{m=1}^M w_m \nabla_{\boldsymbol{\mu}_{t-1}} \mathcal{F}(\boldsymbol{\mu}_{t-1}, e_m) = \sum_{m=1}^M w_m \nabla_{\boldsymbol{\mu}_{t-1}} \mathbb{E}_{\mathcal{N}(\boldsymbol{x}_{t-1};\boldsymbol{\mu}_{t-1},\beta_t \boldsymbol{I})}[f_m(\boldsymbol{x}_{t-1})].
$$

Then, we approximate each component $\nabla_{\boldsymbol{\mu}_{t-1}} \mathbb{E}_{\boldsymbol{x}_{t-1}\sim\mathcal{N}(\boldsymbol{\mu}_{t-1},\beta_t \boldsymbol{I})}[f_m(\boldsymbol{x}_{t-1})]$ using exemplar-wise gradient $\nabla_{c_{t-1}^m} \mathbb{E}_{\boldsymbol{x}_{t-1}\sim\mathcal{N}(c_{t-1}^m,\beta_t \boldsymbol{I})}[f_m(\boldsymbol{x}_{t-1})]$, under the assumption that $\|\delta_m\|$ is small, $m = 1, 2, \ldots, M$.

Therefore, we expand $\mathcal{F}(\boldsymbol{\mu}_{t-1}, e_m) = \mathbb{E}_{\boldsymbol{x}_{t-1}\sim\mathcal{N}(\boldsymbol{\mu}_{t-1},\beta_t \boldsymbol{I})}[f_m(\boldsymbol{x}_{t-1})]$ around $c_{t-1}^m$ using Taylor expansion:

$$
\mathcal{F}(\boldsymbol{\mu}_{t-1}, e_m) = \mathcal{F}(c_{t-1}^m + \delta_m, e_m) = \mathcal{F}(c_{t-1}^m, e_m) + \nabla_{\boldsymbol{\mu}_{t-1}} \mathcal{F}(\boldsymbol{\mu}_{t-1}, e_m)^\top \Big|_{\boldsymbol{\mu}_{t-1}=c_{t-1}^m} \delta_m
$$

$$
+ \frac{1}{2}\delta_m^\top \nabla_{\boldsymbol{\mu}_{t-1}}^2 \mathcal{F}(\boldsymbol{\mu}_{t-1}, e_m)\Big|_{\boldsymbol{\mu}_{t-1}=c_{t-1}^m} \delta_m + \mathcal{O}(\|\delta_m\|^3).
$$

Taking the gradient of both sides with respect to $\boldsymbol{\mu}_{t-1}$ gives:

$$
\nabla_{\boldsymbol{\mu}_{t-1}, e_m} \mathcal{F}(\boldsymbol{\mu}_{t-1}, e_m) = \nabla_{c_{t-1}^m} \mathcal{F}(c_{t-1}^m, e_m) + \nabla_{c_{t-1}^m}^2 \mathcal{F}(c_{t-1}^m, e_m)\delta_m + \mathcal{O}(\|\delta_m\|^2).
$$

Summing over all $m$, the weighted gradient becomes:

$$
\sum_{m=1}^M w_m \nabla_{\boldsymbol{\mu}_{t-1}} \mathcal{F}(\boldsymbol{\mu}_{t-1}, e_m) = \sum_{m=1}^M w_m \left( \nabla_{c_{t-1}^m} \mathcal{F}(c_{t-1}^m, e_m) + \nabla_{c_{t-1}^m}^2 \mathcal{F}(c_{t-1}^m, e_m)\delta_m + \mathcal{O}(\|\delta_m\|^2) \right)
$$

$$
(12)
$$

where $\delta_m = \boldsymbol{\mu}_{t-1} - c_{t-1}^m \ \forall m = 1, 2, \ldots, M$, and we complete the proof.    $\square$

A.3 PROOF OF PROPOSITION 4

**Proposition 4** Given two Gaussian distributions $P = \mathcal{N}(\boldsymbol{c}_{t-1}^m, \beta_t \boldsymbol{I})$ and $Q = \mathcal{N}(\boldsymbol{\mu}_{t-1}, \beta_t \boldsymbol{I})$. Suppose the Euclidean distance between the means is bounded as $\|\boldsymbol{c}_{t-1}^m - \boldsymbol{\mu}_{t-1}\| \leq \varepsilon$. Then, the total variation distance (TV) between the product distributions of $N$ independent samples satisfies $\mathrm{TV}(P^{\otimes N}, Q^{\otimes N}) \leq \frac{N\varepsilon}{\sqrt{4\beta_t}}$.

*Proof.* In the following, we first compute the KL divergence between $P$ and $Q$, which, for Gaussian distributions with the same covariance $\beta_t \boldsymbol{I}$, depends only on the distance between their means. We then apply Pinsker's inequality Fedotov et al. (2003) to bound their total variation distance.

First of all, since both $P$ and $Q$ are Gaussian distributions with identical covariance matrices, their Kullback–Leibler (KL) divergence has a closed-form expression:

$$\mathrm{KL}(P\|Q) = \frac{1}{2\beta_t}\|\boldsymbol{c}_{t-1}^m - \boldsymbol{\mu}_{t-1}\|^2 \leq \frac{\varepsilon^2}{2\beta_t}.$$

By Pinsker's inequality, which relates KL divergence to total variation, we have:

$$\mathrm{TV}(P, Q) \leq \sqrt{\frac{1}{2}\mathrm{KL}(P\|Q)} \leq \frac{\varepsilon}{\sqrt{4\beta_t}}.$$

For the product distributions over $N$ independent samples, the total variation distance satisfies the following inequality:

$$\mathrm{TV}(P^{\otimes N}, Q^{\otimes N}) \leq N \cdot \mathrm{TV}(P, Q) \leq \frac{N\varepsilon}{\sqrt{4\beta_t}}.$$

This implies that the statistical distance between $N$ samples drawn from $P$ and $N$ samples drawn from $Q$ is at most $\frac{N\varepsilon}{\sqrt{4\beta_t}}$. $\square$

**Sample Reward Soup (SRSoup):** Given a set of exemplars $\{\boldsymbol{c}_{t-1}^m\}_{m=1}^M$ introduced in Definition 2, let $\{\boldsymbol{x}_{t-1}^{m,n}\}_{n=1}^N$ denote a set of samples drawn from the exemplar distribution $\mathcal{N}(\boldsymbol{c}_{t-1}^m, \beta_t \boldsymbol{I})$ and $\{f_m(\boldsymbol{x}_{t-1}^{m,n})\}_{n=1}^N$ be the corresponding rewards.

For each exemplar $\boldsymbol{c}_{t-1}^m$, we have

$$\boldsymbol{z}^n = \boldsymbol{x}_{t-1}^{m,n} - \boldsymbol{c}_{t-1}^m, \text{ where } \boldsymbol{x}_{t-1}^{m,n} \sim \mathcal{N}(\boldsymbol{c}_{t-1}^m, \beta_t \boldsymbol{I}),$$

where $\forall n = 1, 2, \ldots, N$.

Following Proposition 4, these samples $\{\boldsymbol{x}_{t-1}^{m,n}\}_{n=1}^N$ can be also considered as sampling $\mathcal{N}(\boldsymbol{\mu}_{t-1}, \beta_t \boldsymbol{I})$ when $\frac{\varepsilon}{\sqrt{\beta_t}}$ is sufficient small. So, we have

$$\bar{\boldsymbol{z}}^n = \boldsymbol{x}_{t-1}^{m,n} - \boldsymbol{\mu}_{t-1} = \boldsymbol{z}^n + \boldsymbol{c}_{t-1}^m - \boldsymbol{\mu}_{t-1}, \text{ where } \boldsymbol{x}_{t-1}^{m,n} \sim \mathcal{N}(\boldsymbol{\mu}_{t-1}, \beta_t \boldsymbol{I}),$$

where $\forall n = 1, 2, \ldots, N$.

Then, these samples can be leveraged for reward-guided search gradient approximation of each component $\nabla_{\boldsymbol{\mu}_{t-1}}\mathcal{F}(\boldsymbol{\mu}_{t-1}, e_m)$ in Equation 3,

$$\begin{aligned}
\nabla_{\boldsymbol{\mu}_{t-1}}\mathcal{F}(\boldsymbol{\mu}_{t-1}, e_m) &\approx \frac{1}{\sqrt{\beta_t}N}\sum_{n=1}^N f_m(\boldsymbol{x}_{t-1}^{m,n})\bar{\boldsymbol{z}}^n \\
&= \frac{1}{\sqrt{\beta_t}N}\sum_{n=1}^N f_m(\boldsymbol{x}_{t-1}^{m,n})(\boldsymbol{z}^n + (\boldsymbol{c}_{t-1}^m - \boldsymbol{\mu}_{t-1})) \\
&= \nabla_{\boldsymbol{c}_{t-1}^m}\mathcal{F}(\boldsymbol{c}_{t-1}^m, e_m) + \frac{1}{\sqrt{\beta_t}N}\sum_{n=1}^N f(\boldsymbol{x}_{t-1}^{m,n})(\boldsymbol{c}_{t-1}^m - \boldsymbol{\mu}_{t-1}),
\end{aligned}$$

## B  EMPIRICAL VERIFICATION USING BHATTACHARYYA COEFFICIENT (BC)

Following Proposition 4, to empirically demonstrate that samples from the exemplar distributions can be reused for other denoising distributions, we adopt Bhattacharyya Coefficient (BC) Bhattacharyya (1943) to measure the overlap between two distributions. BC is a normalized value in the range $[0, 1]$, making it convenient for reference across different image generation cases. Given two continuous probability density functions $p(x)$ and $q(x)$, BC is defined as

$$\text{BC}(p, q) = \int \sqrt{p(x)\, q(x)}\, dx. \tag{13}$$

A higher BC value indicates greater similarity (i.e., more overlap) between the two distributions. Note that the total variation distance (TV) between two distributions is upper bounded by the BC (Sason & Verdú, 2016), i.e., $\text{TV}(p, q) \le \sqrt{2(1 - \text{BC}(p, q)}$.

In the context of diffusion models, the sampling distribution is isotropic, and each dimension of $\boldsymbol{x}_{t-1}$ follows an independent Gaussian distribution with identical covariance $\beta_t$ at each time step $t$. The Bhattacharyya coefficient can be calculated dimension-wise as:

$$\text{BC}(\boldsymbol{c}^m_{d,t-1}, \boldsymbol{\mu}_{d,t-1}) = \exp\left(-\frac{1}{8\beta_t}\|\boldsymbol{\mu}_{d,t-1} - \boldsymbol{c}^m_{d,t-1}\|^2\right), \tag{14}$$

where $\boldsymbol{\mu}_{d,t-1}$ and $\boldsymbol{c}^m_{d,t-1}$ denote the $d$-th dimensions of $\boldsymbol{\mu}_{t-1}$ and $\boldsymbol{c}^m_{t-1}$, respectively.

We plot BC values for $\boldsymbol{\mu}_{t-1}$ and $\boldsymbol{c}^m_{t-1}$ in Fig. 8 under the setting of optimizing the aesthetic and compressiblity rewards using the true black-box gradient of weighted sum objectives. Note that the weights for $\boldsymbol{c}^1_{t-1}, \boldsymbol{c}^2_{t-1}$ and are $[1, 0], [0, 1], [0.5, 0.5]$, respectively. In Fig. 8, at the beginning, we have $\text{BC}(\boldsymbol{c}^m_{d,T-1}, \boldsymbol{\mu}_{d,T-1}) = 1$ because $\boldsymbol{\mu}_{T-1} = \boldsymbol{c}^1_{T-1} = \ldots = \boldsymbol{c}^M_{T-1}$, resulting from initialization with the same noise $\boldsymbol{x}_T \sim \mathcal{N}(\mathbf{0}, \boldsymbol{I})$. At the early stage of denoising process (i.e., $t > 30$), more than $50\%$ of the dimensions have $\text{BC}(\boldsymbol{c}^m_{d,t-1}, \boldsymbol{\mu}_{d,t-1}) > 0.4$. which suggests that our SRSoup provides a reliable approximation. At the later stage, this term approaches zero, i.e., $\text{BC}(\boldsymbol{c}^m_{d,t-1}, \boldsymbol{\mu}_{d,t-1}) \xrightarrow{t \to 0} 0$, making the approximation ineffective.

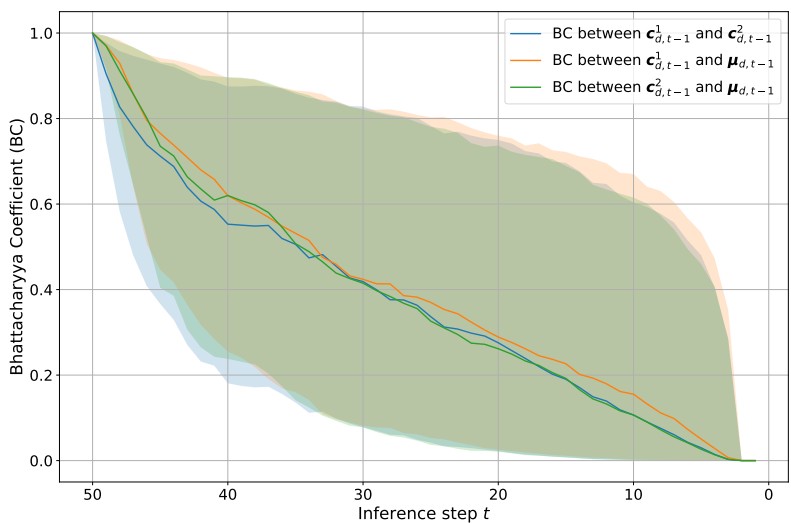

Figure 8: The Bhattacharyya coefficient (BC) is calculated between any two of the following denoised distributions, i.e., $\mathcal{N}(\boldsymbol{x}_{t-1}; \boldsymbol{\mu}_{t-1}, \beta_t\boldsymbol{I})$, $\mathcal{N}(\boldsymbol{x}_{t-1}; \boldsymbol{c}^1_{t-1}, \beta_t\boldsymbol{I})$ and $\mathcal{N}(\boldsymbol{x}_{t-1}; \boldsymbol{c}^2_{t-1}, \beta_t\boldsymbol{I})$. In the legend, they are simplified as "BC between distribution mean * and *", respectively. Since BC is calculated dimension-wise, the solid line represents the median BC across all dimensions, while the shaded area covers the range between the 25th and 75th percentiles. Note $\mathcal{N}(\boldsymbol{x}_{t-1}; \boldsymbol{c}^1_{t-1}, \beta_t\boldsymbol{I})$ optimizes the aesthetic reward function $f_1$, $\mathcal{N}(\boldsymbol{x}_{t-1}; \boldsymbol{c}^2_{t-1}, \beta_t\boldsymbol{I})$ optimizes the compressibility function $f_2$ and $\mathcal{N}(\boldsymbol{x}_{t-1}; \boldsymbol{\mu}_{t-1}, \beta_t\boldsymbol{I})$ optimizes $0.5f_1 + 0.5f_2$.

Table 1: Cross-reward evaluation for the case of aligning with the aesthetic & compressibility reward $w_1 f_1 + w_2 f_2$ on animal prompts.

| Cross-reward | Method | $w_1 = 0$ | $w_1 = 0.2$ | $w_1 = 0.4$ | $w_1 = 0.6$ | $w_1 = 0.8$ | $w_1 = 1$ |
|---|---|---|---|---|---|---|---|
| CLIP (SD1.5: 0.2589) | SRSoup | 0.2692 | 0.2672 | 0.2735 | 0.2690 | 0.2549 | 0.2532 |
| | AlignProp (soup) | 0.1844 | 0.2298 | 0.2650 | 0.2446 | 0.2527 | 0.2464 |
| | TDPO (soup) | 0.2379 | 0.2628 | 0.2553 | 0.2403 | 0.2590 | 0.2361 |
| | DDPO (soup) | 0.2583 | 0.2665 | 0.2582 | 0.2561 | 0.2573 | 0.2552 |
| HPSv2 (SD1.5: 0.2731) | SRSoup | 0.2401 | 0.2423 | 0.2581 | 0.2676 | 0.2680 | 0.2673 |
| | AlignProp (soup) | 0.2189 | 0.2338 | 0.2571 | 0.2708 | 0.2669 | 0.2637 |
| | TDPO (soup) | 0.2372 | 0.2538 | 0.2629 | 0.2638 | 0.2556 | 0.2514 |
| | DDPO (soup) | 0.2440 | 0.2562 | 0.2644 | 0.2681 | 0.2687 | 0.2621 |

This justifies our hybrid strategy for multi-reward guidance. At the early stage, when the distribution $\mathcal{N}(\boldsymbol{\mu}_{t-1}, \beta_t \boldsymbol{I})$ exhibits significant overlap with the exemplar distributions $\mathcal{N}(\boldsymbol{c}_{t-1}^m, \beta_t \boldsymbol{I})$ $\forall m = 1, 2, \ldots, M$, we perform Sample Reward Soups (Algorithm 2). At the later stage, when the distribution $\mathcal{N}(\boldsymbol{\mu}_{t-1}, \beta_t \boldsymbol{I})$ becomes distinct from the exemplar distributions, we adopt the true update guided by weighted rewards (Equation 9).

## C   ADDITIONAL EXPERIMENT RESULTS

### C.1   CROSS-REWARD EVALUATION

We include the cross-evaluation metrics for the three scenarios in Fig. 3. Table 1, Table 2 and Table 3 show the evaluation using the CLIP score, which measures prompt alignment and the evaluation using other unseen reward models during alignment.

The evaluation results for the pretrained Stable Diffusion 1.5 (SD1.5) can be taken as the reference. From the above tables, we can find that: (1) AlignProp (soup) has a significant drop (denoted in red color) in the cross-evaluation metrics compared to SD1.5 when aligning with the aesthetic and compressibility rewards, which indicates reward over-optimization, consistent with the visual results shown in Fig. 4. (2) TDPO (soup) also suffers from the similar issue when optimizing the aesthetic reward on HPD prompts (Fig. 14). (3) When aligning with the aesthetic reward ($w_1 = 1$), we observe some drops in the CLIP metric compared to SD1.5 across all methods, possibly indicating a loss of prompt alignment. But our SRSoup still achieves better results than other baselines.

Table 2: Cross-reward evaluation for the case of aligning with the aesthetic & HPSv2 reward $w_1 f_1 + w_2 f_2$ on animal prompts.

| Cross-reward | Method | $w_1 = 0$ | $w_1 = 0.2$ | $w_1 = 0.4$ | $w_1 = 0.6$ | $w_1 = 0.8$ | $w_1 = 1$ |
|---|---|---|---|---|---|---|---|
| CLIP (SD1.5: 0.2574) | SRSoup | 0.2571 | 0.2566 | 0.2591 | 0.2545 | 0.2445 | 0.2324 |
| | AlignProp (soup) | 0.2627 | 0.2646 | 0.2635 | 0.2631 | 0.2477 | 0.2171 |
| | TDPO (soup) | 0.2542 | 0.2613 | 0.2662 | 0.2583 | 0.2110 | 0.1128 |
| | DDPO (soup) | 0.2579 | 0.2568 | 0.2516 | 0.2509 | 0.2490 | 0.2237 |
| PickScore (SD1.5: 0.2190) | SRSoup | 0.2300 | 0.2291 | 0.2277 | 0.2236 | 0.2189 | 0.2142 |
| | AlignProp (soup) | 0.2198 | 0.2216 | 0.2213 | 0.2214 | 0.2162 | 0.2069 |
| | TDPO (soup) | 0.2198 | 0.2221 | 0.2218 | 0.2201 | 0.2056 | 0.1827 |
| | DDPO (soup) | 0.2219 | 0.2235 | 0.2242 | 0.2229 | 0.2163 | 0.2065 |

### C.2   SRSOUP APPLIED IN SD3 AND SDXL

Our SRSoup method also generalizes to other T2I diffusion models, such as SDXL, and flow models based on diffusion transformers (DiT), such as Stable Diffusion 3.5 Medium (SD3) (Esser et al., 2024). Fig. 9(d) shows that SRSoup improves both the aesthetic reward and the PickScore reward compared to the pretrained SD3, and achieves a trade-off between the two rewards.

Table 3: Cross-reward evaluation for the case of aligning with the aesthetic & PickScore reward $w_1 f_1 + w_2 f_2$ on animal prompts.

| Cross-reward | Method | $w_1 = 0$ | $w_1 = 0.2$ | $w_1 = 0.4$ | $w_1 = 0.6$ | $w_1 = 0.8$ | $w_1 = 1$ |
|---|---|---|---|---|---|---|---|
| CLIP (SD1.5: 0.2574) | SRSoup | 0.2625 | 0.2594 | 0.2562 | 0.2543 | 0.2451 | 0.2238 |
| | AlignProp (soup) | 0.2404 | 0.2423 | 0.2500 | 0.2560 | 0.2460 | 0.2171 |
| | TDPO (soup) | 0.2468 | 0.2522 | 0.2548 | 0.2526 | 0.2063 | 0.1128 |
| | DDPO (soup) | 0.2439 | 0.2460 | 0.2475 | 0.2524 | 0.2449 | 0.2237 |
| HPSv2 (SD1.5: 0.2787) | SRSoup | 0.2878 | 0.2872 | 0.2860 | 0.2833 | 0.2762 | 0.2714 |
| | AlignProp (soup) | 0.2785 | 0.2793 | 0.2797 | 0.2790 | 0.2747 | 0.2655 |
| | TDPO (soup) | 0.2742 | 0.2803 | 0.2794 | 0.2783 | 0.2662 | 0.2493 |
| | DDPO (soup) | 0.2826 | 0.2829 | 0.2820 | 0.2796 | 0.2735 | 0.2647 |

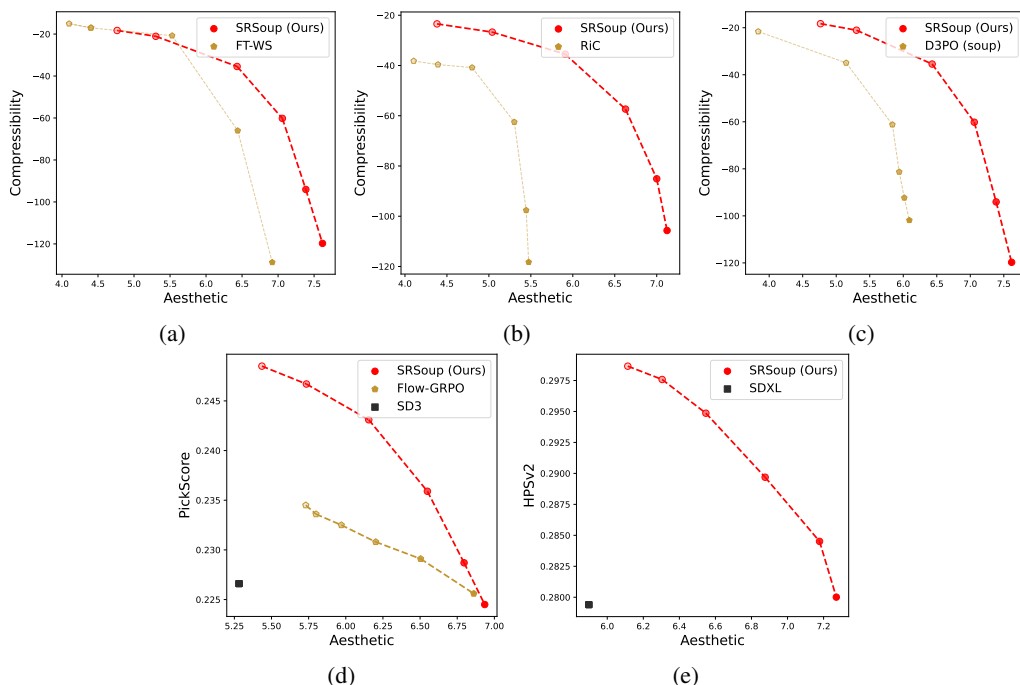

Figure 9: (a) SRSoup with the SD1.5 base vs. fine-tuning with weighted-sum objectives. (b) SRSoup with the SD1.5 base vs. RiC. (c)SRSoup with the SD1.5 base vs. D3PO. (d) SRSoup with the SD3 base vs. Flow-GRPO. (e) Pareto front of SRSoup with the SDXL base, aligned on two rewards.

Fig. 9(b) shows that SRSoup improves both the aesthetic reward and the HPSv2 reward compared to the pretrained SDXL, and achieves a trade-off between the two rewards. Visual results are provided in Fig.15. Compared to the pretrained SDXL, SRSoup improves both the visual aesthetics and the prompt consistency.

## C.3 COMPARED WITH OTHER FINE-TUNING METHODS

DPO methods (e.g., DiffusionDPO (Wallace et al., 2024), D3PO (Yang et al., 2024a)) are designed to guide T2I models using human preferences, which is not directly applicable to the setting of guiding T2I models using multi-reward models studied in this paper. To make a comparison, we follow D3PO, which adopts reward models for providing preference pairs. Specifically, the aesthetic reward model and compressibility reward model are used to provide preference pairs for fine-tuning SD1.5 with the DPO loss, respectively. Then similarly to DDPO (soup), we interpolate the two finetuned models for obtaining multi-reward guided samples. Fig. 6(c) and Fig. 12 demonstrates that our SRSoup achieves better performance than D3PO (soup).

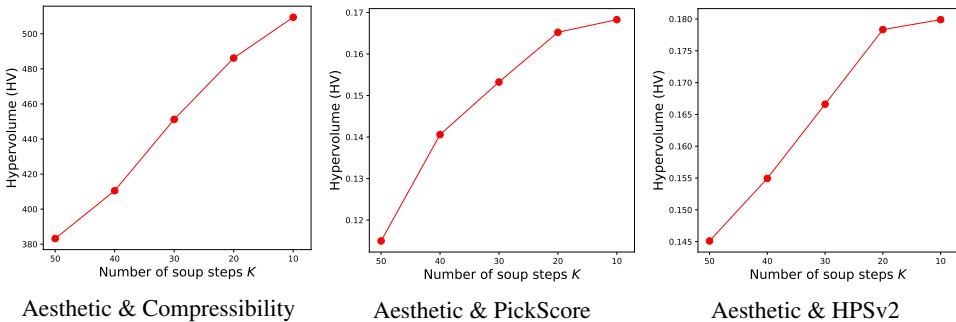

Figure 10: Hypervolumn (HV) versus different soup steps $K$.

Flow-GRPO (Liu et al., 2025) was recently proposed for reward alignment using GRPO. To compare with it, we apply Flow-GRPO to align the pretrained SD3 model with the aesthetic and PickScore rewards on the HPD prompts, respectively, to obtain fine-tuned models. We then apply the soup strategy to generate multi-reward–aligned samples. Fig. 9(d) shows that SRSoup outperforms Flow-GRPO in terms of multi-reward alignment. Meanwhile, the visual results in Fig. 13 demonstrate that SRSoup achieves better prompt alignment.

## C.4 COMPARED WITH MULTI-REWARD FINE-TUNING METHODS

As discussed in Section 2, multi-reward training methods have some drawbacks. Fine-tuning on weighted sum requires training a large number of models to trade off different reward functions. Parrot (Lee et al., 2024) and Rewards-in-context (RiC) (Yang et al., 2024b) rely on a large supervised dataset. We include comparisons with fine-tuning using weighted sum objectives (FT-WS) and RiC, as shown in Fig. 9. Since Parrot's code has not been released, we do not compare against it.

SRSoup vs. FT-WS. Fig. 9(a) shows the results of SRSoup and FT-WS when aligning with the aesthetic and compressibility rewards ($w_1 f_1 + w_2 f_2$) on the animal prompts. The depth of the marker color denotes the value of $w_1$; a deeper color indicates a larger value. The values of $w_1$ are set to 0.0, 0.2, 0.4, 0.6, 0.8, and 1.0, respectively. The result demonstrates that SRSoup outperforms FT-WS.

SRSoup vs. RiC. For an ease of comparison, we follow the text-to-image setting in RiC to conduct the comparison experiments, which aligns with the aesthetic and compressibility rewards ($w_1 f_1 + w_2 f_2$) on the CoCo dataset. Fig. 9(a) shows that SRSoup outperforms RiC.

## C.5 HARD PROMPTS VS. EASY PROMPTS

We find that different prompt samples exhibit varying levels of optimization difficulty. As shown in Fig. 16, with a query number of $N = 20$, the prompt "snail" is optimized effectively, while "hippopotamus" is not when optimizing the compressibility reward. As $N$ increases, the optimization of "hippopotamus" improves. This is reasonable, as the concept of a "snail" is simpler than that of a "hippopotamus". As discussed in the limitations of the main text, future work could focus on developing an adaptive query allocation strategy based on the optimization difficulty of individual samples under a fixed query budget.

## C.6 COMPARISON OF REWARD QUERIES AND INFERENCE TIME

As elaborated in the main text, the update guided by weighted rewards (Equation 9, a.k.a., weighted sum) requires $NTM(L - M + 1)$ per prompt. Our SRSoup would require $NTM(L - M + 1) - NKM(L - M)$. The reduction in reward queries is $NKM(L - M)$.

In Table 4, we compare our SRSoup under different settings of $K$ to the weighted sum method in terms of the required reward queries and inference time for generating a single image. For SRSoup with $K = 20$, which achieves performance relatively close to that of the weighted sum method (Fig. 7 in the main text), the reduction in reward queries reaches 4800. In fact, the reduction can be around 40% when a large number of weight vectors are considered ($\lim_{L \to +\infty} \frac{NKM(L-M)}{NTM(L-M+1)} = \frac{K}{T} = 40\%$).

We plot the HV in terms of $K$ in Fig. 10 and present GPU memory and inference time breakdown in Table 5 and Table 4, respectively.

Table 4: The total number of reward queries and inference time required for one prompt under different setting of $K$ while optimizing the aesthetic reward $f_1$ and the compressibility reward $f_2$. $N, T, L$ are 30, 50, 6, respectively.

| Method | SRSoup | | | | | WeightedSum |
|---|---|---|---|---|---|---|
| | $K = 50$ | $K = 40$ | $K = 30$ | $K = 20$ | $K = 10$ | |
| #Query | 3,000 | 5,400 | 7,800 | 10,200 | 12,600 | 15,000 |
| Time (min) | $9.7 \pm 0.1$ | $13.7 \pm 0.3$ | $17.3 \pm 0.2$ | $21.1 \pm 0.2$ | $24.9 \pm 0.2$ | $28.8 \pm 0.3$ |

Table 5: The peak GPU memory required for SRSoup and WeigthedSum (WS) when optimizing aesthetic & compressiblity rewards on the animal prompts and when optimizing aesthetic & PickScore rewards and aesthetic & PickScore rewards on the HPD prompts in Fig. 1. See Section D for detailed setup. The difference of required GPU memory is due to the different setup of $N$.

| | SRSoup | WS($N = 26$) | WS($N = 21$) | WS($N = 16$) | WS($N = 11$) | WS($N = 6$) |
|---|---|---|---|---|---|---|
| animal | 20.9GB | 20.9GB | 20.9GB | 20.9GB | 20.9GB | 18.4GB |
| HPD | 24.6GB | 24.6GB | 24.6GB | 24.6GB | 24.6GB | 22.1GB |

Table 6: Inference-time (minute) breakdown required for one prompt under different numbers of reward queries. *Total* refers to the total time for full pipeline inference. *Decode* refers to time spent decoding latent embeddings back into images for reward evaluation; *Reward* refers to time spent evaluating decoded images with reward models.

| Method | #Queries | Aethetic & Compressibility | | | Aesthetic & PickScore | | | Aesthetic & HPSv2 | | |
|---|---|---|---|---|---|---|---|---|---|---|
| | | Total | Decode | Reward | Total | Decode | Reward | Total | Decode | Reward |
| SRSOUP | 3000 | 9.2 | 8.8 | 0.2 | 9.7 | 9.0 | 0.3 | 9.7 | 9.0 | 0.3 |
| | 5400 | 13.0 | 12.2 | 0.4 | 14.0 | 13.0 | 0.6 | 13.7 | 12.7 | 0.6 |
| | 7800 | 17.1 | 15.9 | 0.6 | 17.4 | 16.2 | 0.8 | 17.6 | 16.2 | 0.9 |
| | 10200 | 20.4 | 19.2 | 0.8 | 21.6 | 20.0 | 1.1 | 21.6 | 20.0 | 1.2 |
| | 12600 | 24.3 | 23.1 | 0.9 | 25.1 | 23.3 | 1.4 | 25.4 | 23.4 | 1.6 |
| WS | 3000 | 8.2 | 7.5 | 0.3 | 9.3 | 8.4 | 0.5 | 9.4 | 8.5 | 0.5 |
| | 5500 | 12.8 | 12.1 | 0.4 | 14.4 | 13.3 | 0.7 | 12.2 | 11.1 | 0.7 |
| | 8000 | 17.7 | 16.8 | 0.6 | 18.4 | 17.0 | 1.0 | 15.7 | 14.3 | 1.0 |
| | 10500 | 21.7 | 20.4 | 0.9 | 23.0 | 21.4 | 1.2 | 21.1 | 19.2 | 1.4 |
| | 13000 | 24.8 | 23.4 | 1.0 | 27.1 | 25.3 | 1.4 | 25.8 | 23.7 | 1.7 |

### C.7 ADDITIONAL RESULTS FOR TWO-OBJECTIVE ALIGNMENT

We present more results for the optimization of the aesthetic and compressibility rewards on animal prompts in Fig. 18, Fig. 19, Fig. 20 and Fig. 21. The images generated from different prompts, optimized for the aesthetic reward by DDPO, TDPO, and AlignProp, exhibit similar backgrounds, indicating over-optimization in these fine-tuning methods. In contrast, our SRSoup produces images with greater diversity.

A qualitative comparison of the optimization of the aesthetic and HPSv2 rewards is shown in Fig. 17. It demonstrates the superiority of our method in prompt alignment. Additionally, the baselines still tend to over-optimize the aesthetic reward on HPD prompts. The results for the the optimization of the aesthetic and PickScore rewards on HPD prompts can be seen in Fig. 22, Fig. 23, Fig. 24 and Fig. 25.

## D ADDITIONAL IMPLEMENTATION DETAILS

**SRSoup.** We run our experiments on NVIDIA RTX 6000 Ada GPU (48GB). Typically, we set the denoising step $T$ as 50 following previous works (Black et al., 2024; Prabhudesai et al., 2023; Zhang

et al., 2024). While more queries often lead to better results, $N = 30$ is sufficient to achieve good performance. The number of soup steps $K$ is set to 20, based on the analysis of BC explained above. In order to coordinate scales of different reward functions, we normalize the reward values using their mean and standard deviation. The reward model trained on clean images may have inaccurate evaluations on noisy images. Thus, we follow a commonly used technique to reduce errors, which performs evaluation on the image denoised from the noisy images (Eq. (11) in (Chung et al., 2023)). For the three-objective experiment in Fig. 5 of the main text, we use weights $\big[[1.0, 0.0, 0.0], [0.0, 1.0, 0.0], [0.0, 0.0, 1.0], [\frac{1}{3}, \frac{1}{3}, \frac{1}{3}], [0.1, 0.1, 0.8], [0.1, 0.8, 0.1], [0.8, 0.1, 0.1], [0.2, 0.2, 0.6], [0.2, 0.6, 0.2], [0.6, 0.2, 0.2], [0.4, 0.4, 0.2], [0.4, 0.2, 0.4], [0.2, 0.4, 0.4]\big]$ to depict the Pareto front.

**Baseline.** We use the official code from DDPO, TDPO, and AlignProp, with an effective batch size of 256 achieved through gradient accumulation. Following their original settings, DDPO and TDPO are fine-tuned for 200 and 100 epochs, respectively, by default. Because they collapse on the animal prompt (generating meaningless images), we evaluated their performance using checkpoints before the collapse. For DDPO, we used the checkpoint at the 130th epoch for the aesthetic reward and the 80th epoch for the compressibility reward. For TDPO, we used the checkpoint at the 80th epoch for the aesthetic reward and the 30th epoch for the compressibility reward. Since AlignProp tends to collapse more easily, we also used pre-collapse checkpoints for comparison. For animal prompts, we early stop at 6 epochs for the aesthetic reward; and 2 epochs for the compressibility reward; for HPD prompts, we early stop at 12 epochs for the aesthetic reward, 2 epochs for the HPSv2 reward and 23 for the PickScore reward. We plot the rewards evaluating on different checkpoints in Fig. 11.

**Query setup in Section 5.3** In Fig. 1, the number of denoising timesteps $T$ is set to 50 for both methods, and the number of weight vectors is 6 in the two-objective cases. For SRSoup, the number of query samples per reward function at each timestep $N$ is fixed at 30, while $K$ is varied to control the overall query budget. Specifically, $K$ is set to $50, 40, 30, 20$, and $10$, corresponding to $3000, 5400, 7800, 10200$, and $12600$ required queries per prompt, respectively. For weighted-sum guidance, to match a similar query budget, $N$ is set to $6, 11, 16, 21$, and $26$, resulting in $3000, 5500, 8000, 10500$, and $13000$ required queries per prompt, respectively. The reference points for calculating HV values in the three cases (aligning with the aesthetic and compressibility reward; with the aesthetic and HPSv2 reward; and with the aesthetic and PickScore reward) are chosen as values slightly lower than the worst value among all obtained evaluation results, namely $(4, -180), (5, 0.23), (5, 0.18)$, respectively.

# E  USAGE OF LARGE LANGUAGE MODEL (LLM)

The LLM (ChatGPT from OpenAI) was applied to refine sentence structures, improve readability, and ensure grammatical correctness, without altering the scientific content, analysis, or conclusions in this paper.

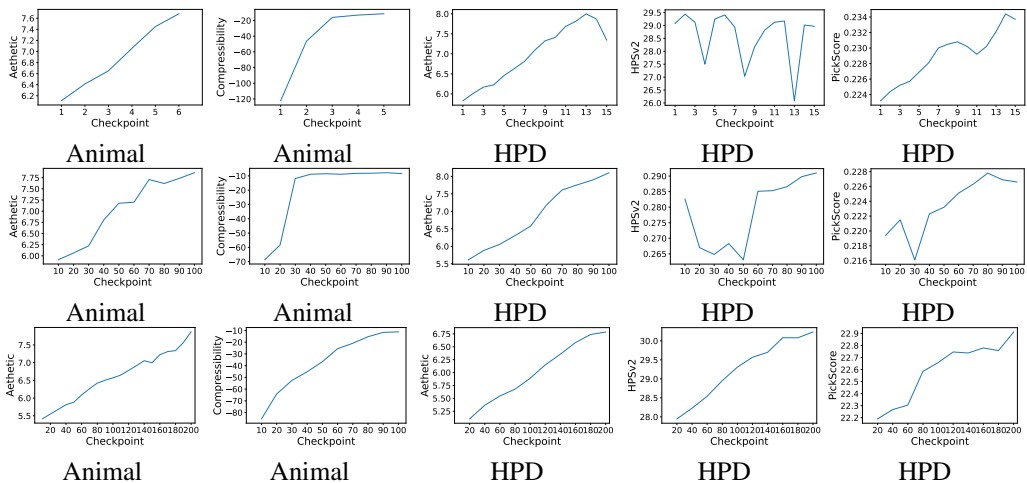

Figure 11: Reward values during the finetuning of AlignProp (first row), TDPO (second row) and DDPO (third row) on the animal prompts and HPD prompts.

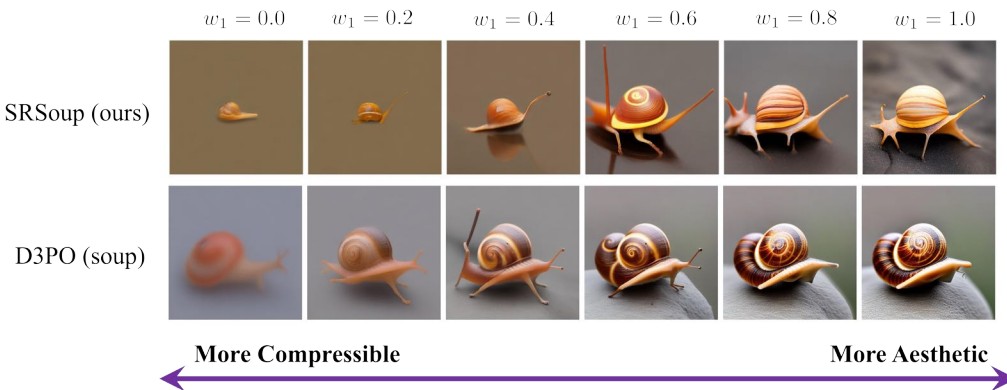

Figure 12: Qualitative comparison of T2I model aligned with the aesthetic reward and the PickScore reward $w_1 f_1 + w_2 f_2$. Prompt: "snail".

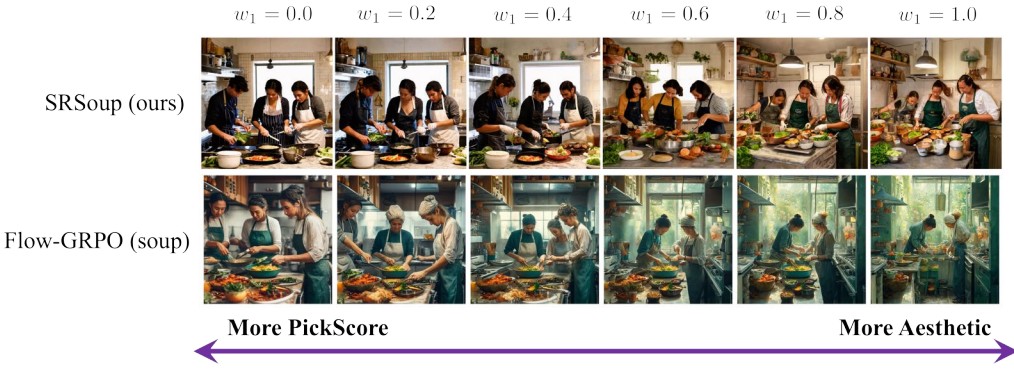

Figure 13: Qualitative comparison of T2I model aligned with the aesthetic reward and the PickScore reward $w_1 f_1 + w_2 f_2$. Prompt: "Three people are preparing a meal in a small kitchen.".

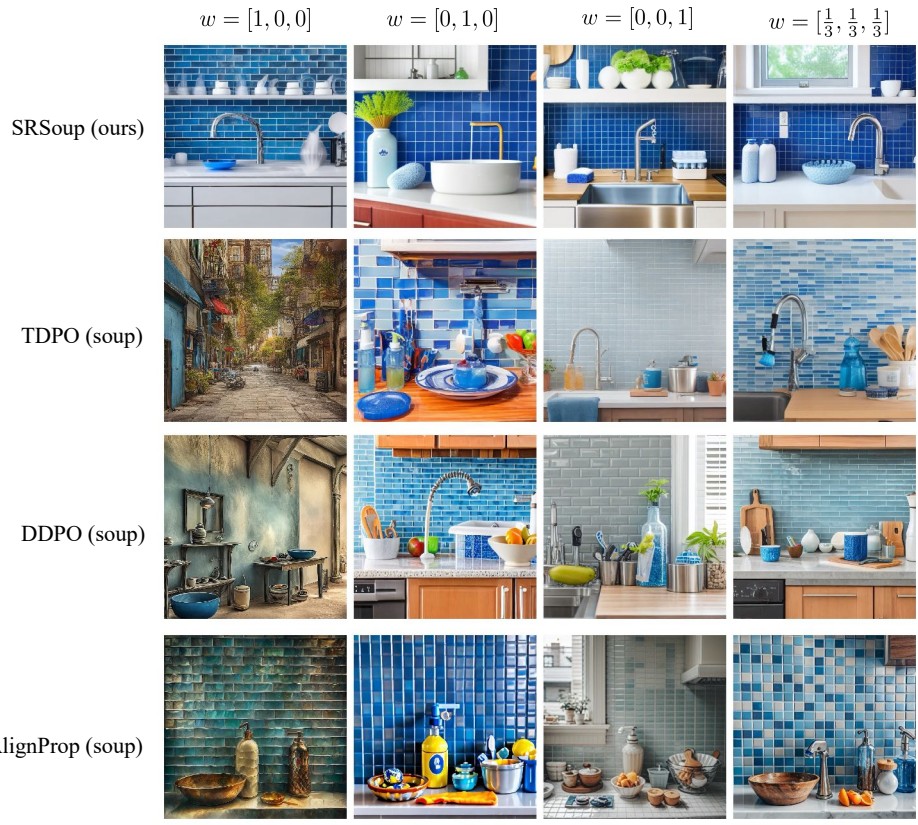

Figure 14: Qualitative comparison of T2I model aligned with the aesthetic reward $f_1$, the HPSv2 reward $f_2$ and the PickScore reward $f_3$. $w$ is used to weight $f_1$, $f_2$ and $f_3$. Prompt: "An eye level counter-view shows blue tile, a faucet, dish scrubbers, bowls, a squirt bottle and similar kitchen items.".

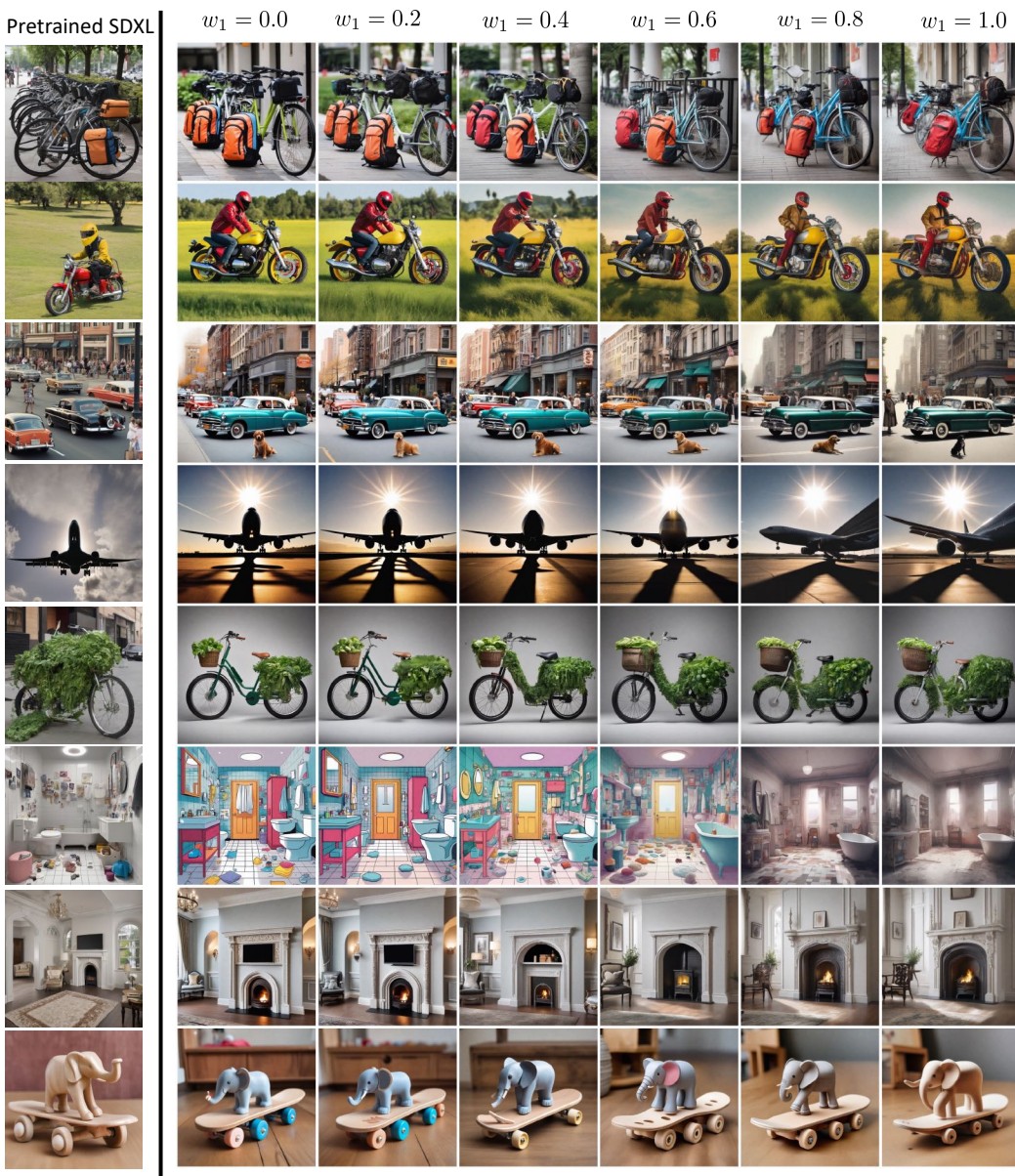

Figure 15: Our SRSoup with SDXL base aligned with the aesthetic reward $f_1$ and HPSv2 reward $f_2$. The weights for $f_1$ and $f_2$ are $w_1$ and $w_2 = 1 - w_1$, respectively. Prompts are "Bicycles with backpacks parked in a public place", "Yellow and red motorcycle with a man riding on it next to grass", "Classic cars on a city street with people and a dog", "A large commercial airliner silhoetted in the sun", "A motorized bicycle covered with greens and beans", "There is a bathroom that has a lot of things on the floor", "Ornate archway inset with matching fireplace in room", "A wooden skate with a toy elephant inside of it".

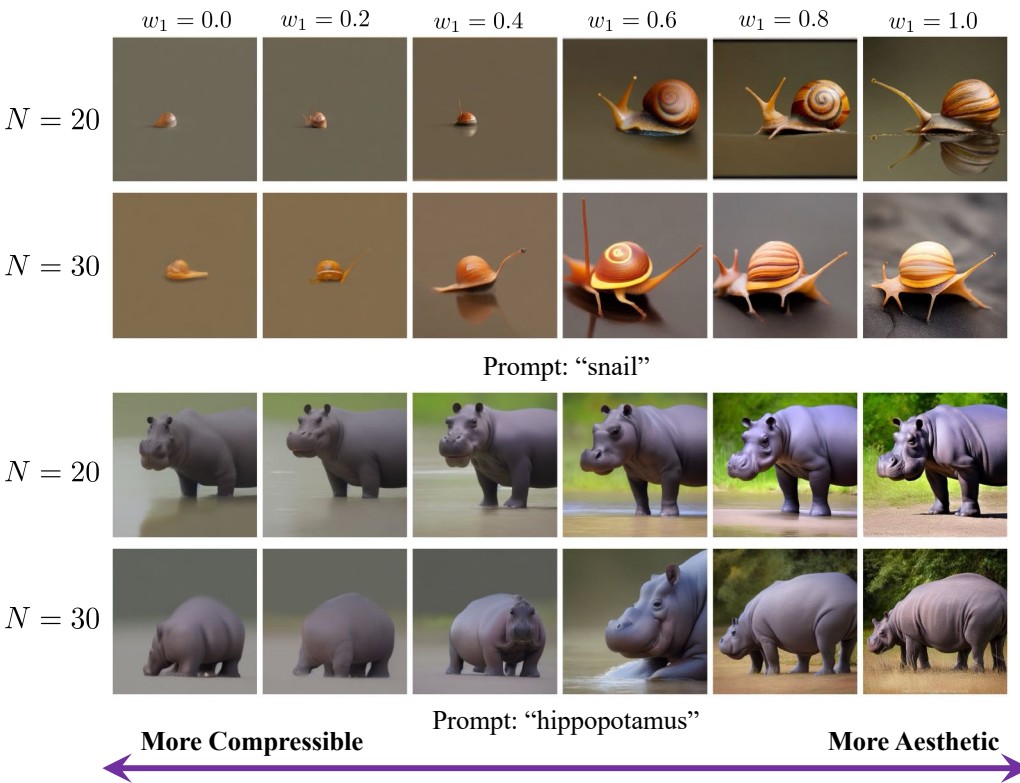

Figure 16: Our SRSoup for different prompts. The target is to optimize the aesthetic reward $f_1$ and compressibility reward $f_2$. The weights for $f_1$ and $f_2$ are $w_1$ and $w_2 = 1 - w_1$, respectively.

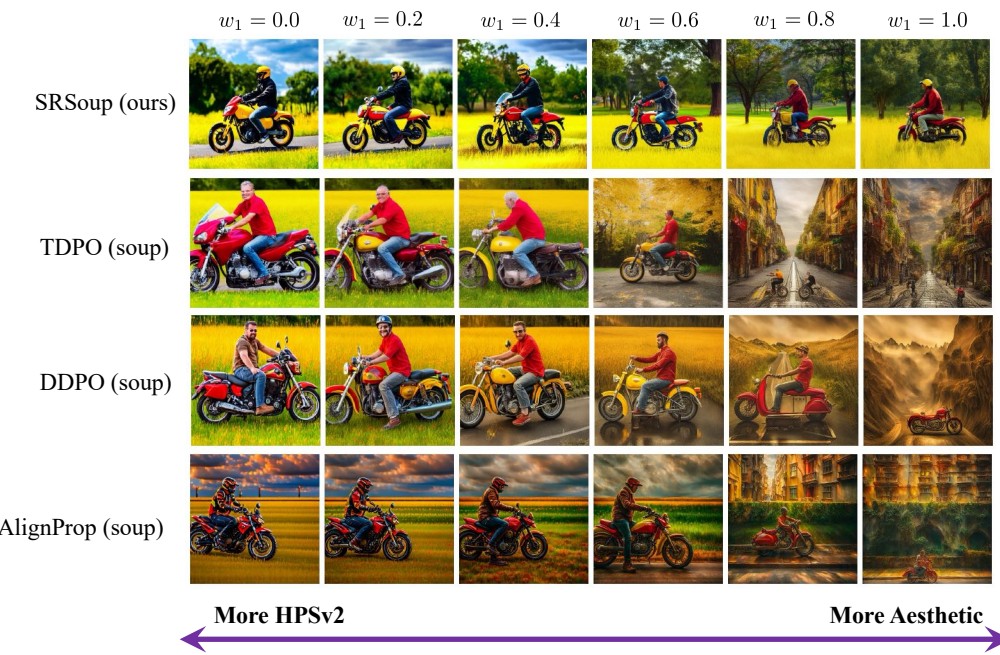

Figure 17: Qualitative comparison of T2I model aligned with the aesthetic reward $f_1$ and HPSv2 reward $f_2$. The weights for $f_1$ and $f_2$ are $w_1$ and $w_2 = 1 - w_1$, respectively. Prompt: "yellow and red motorcycle with a man riding on it next to grass".

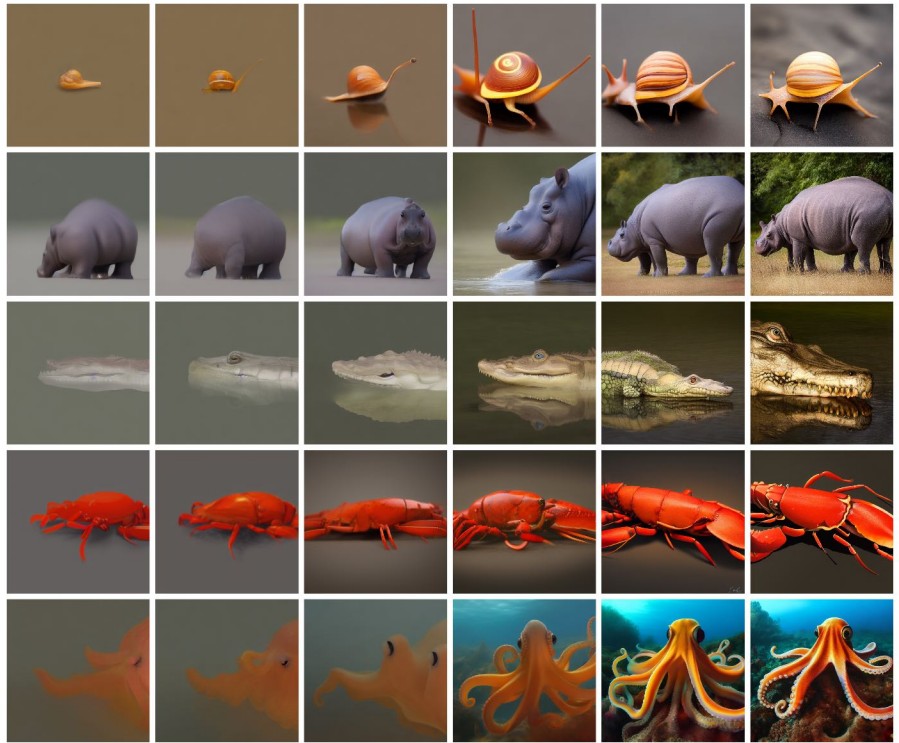

Figure 18: Our SRSoup aligned with the aesthetic reward $f_1$ and compressibility reward $f_2$. The weights for $f_1$ and $f_2$ are $w_1$ and $w_2 = 1 - w_1$, respectively. From left to right, $w_1 = 0.0, 0.2, 0.4, 0.6, 0.8, 1.0$. Prompts: "snail"; "hippopotamus"; "crocodile"; "lobster"; "octopus".

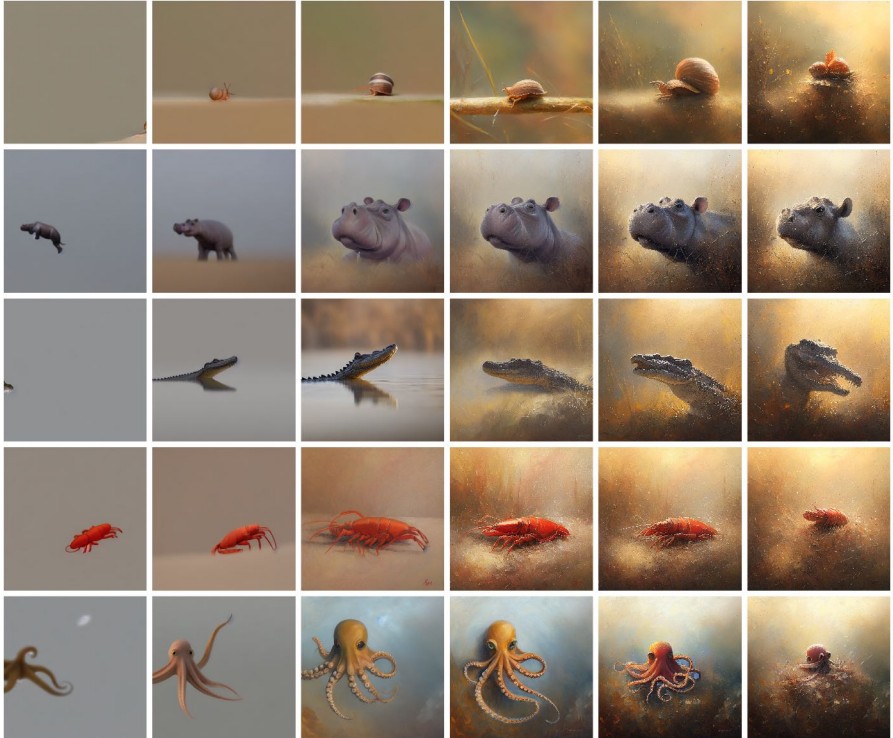

Figure 19: TDPO aligned with the aesthetic reward $f_1$ and compressibility reward $f_2$. The weights for $f_1$ and $f_2$ are $w_1$ and $w_2 = 1 - w_1$, respectively. From left to right, $w_1 = 0.0, 0.2, 0.4, 0.6, 0.8, 1.0$. Prompts: "snail"; "hippopotamus"; "crocodile"; "lobster"; "octopus".

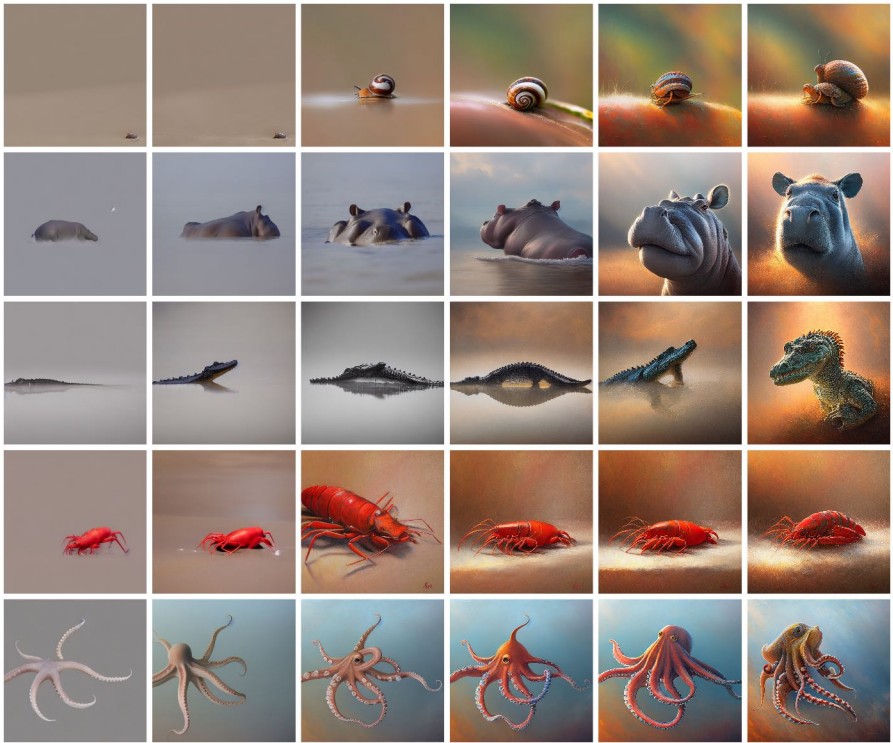

Figure 20: DDPO aligned with the aesthetic reward $f_1$ and compressibility reward $f_2$. The weights for $f_1$ and $f_2$ are $w_1$ and $w_2 = 1 - w_1$, respectively. From left to right, $w_1 = 0.0, 0.2, 0.4, 0.6, 0.8, 1.0$. Prompts: "snail"; "hippopotamus"; "crocodile"; "lobster"; "octopus".

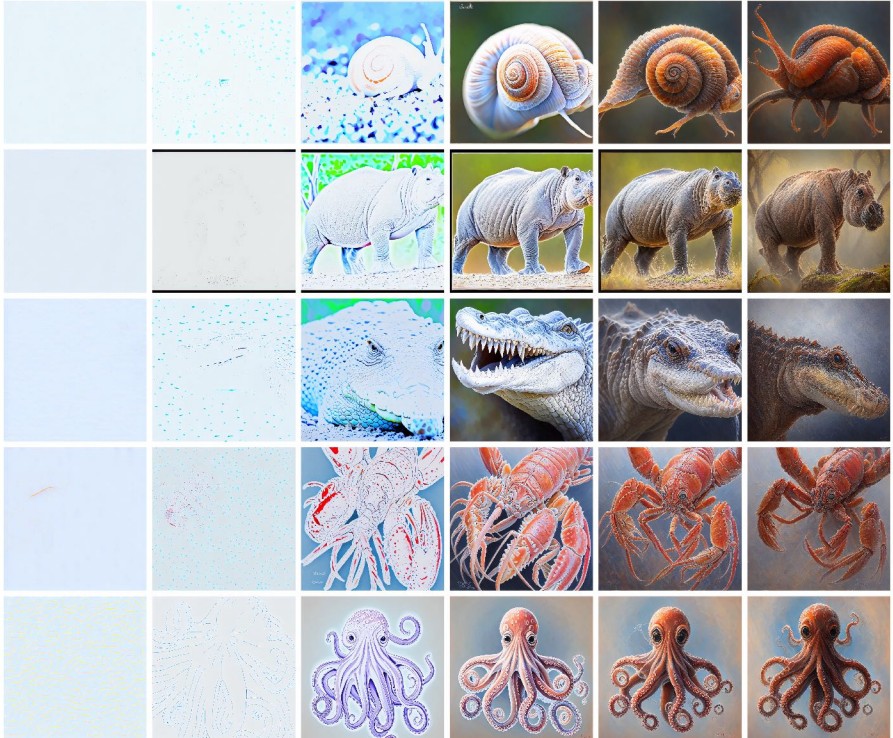

Figure 21: AlignProp aligned with the aesthetic reward $f_1$ and compressibility reward $f_2$. The weights for $f_1$ and $f_2$ are $w_1$ and $w_2 = 1 - w_1$, respectively. From left to right, $w_1 = 0.0, 0.2, 0.4, 0.6, 0.8, 1.0$. Prompts: "snail"; "hippopotamus"; "crocodile"; "lobster"; "octopus".

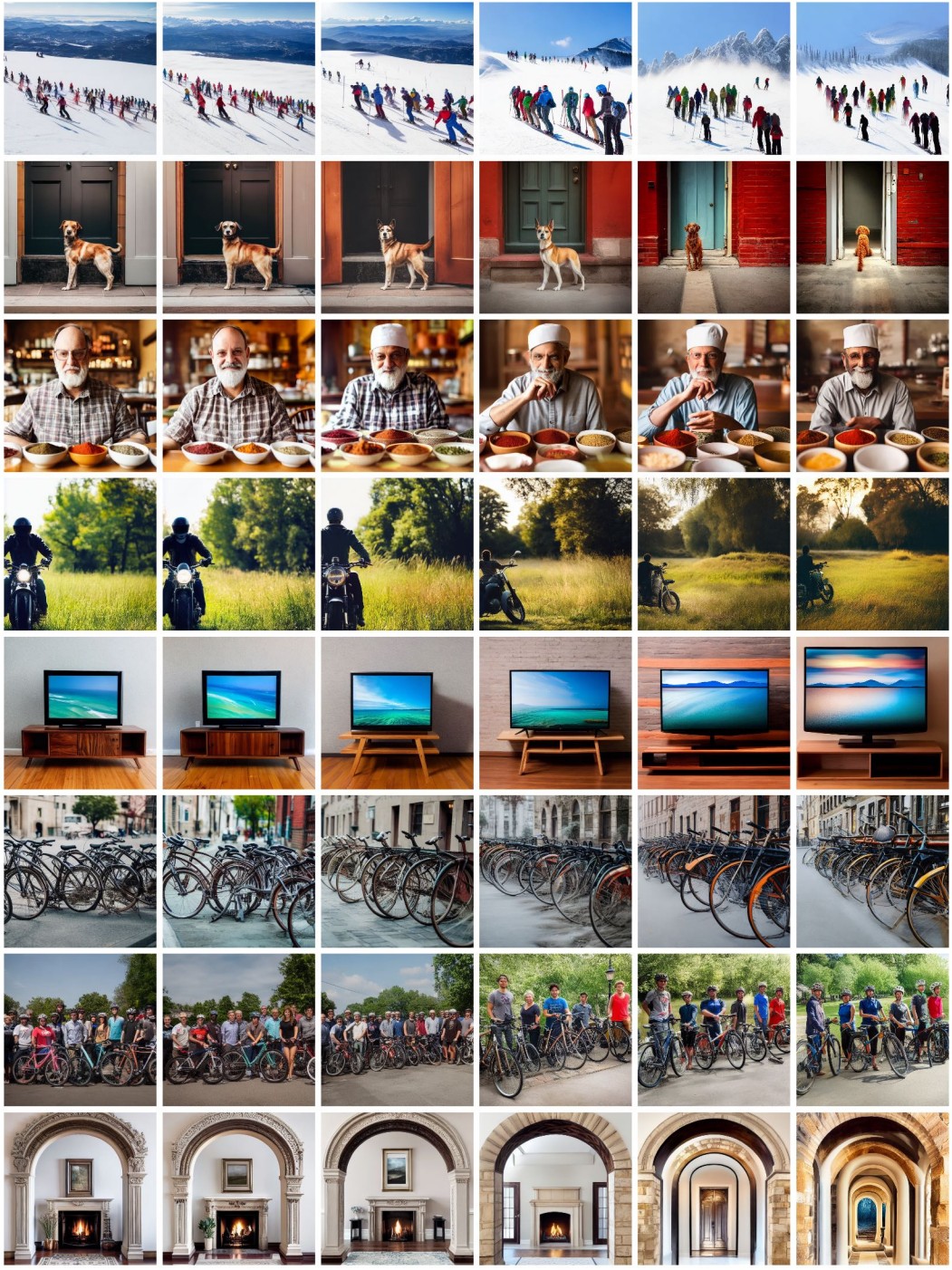

Figure 22: Our SRSoup aligned with the aesthetic reward $f_1$ and PickScore reward $f_2$. The weights for $f_1$ and $f_2$ are $w_1$ and $w_2 = 1 - w_1$, respectively. From left to right, $w_1 = 0.0, 0.2, 0.4, 0.6, 0.8, 1.0$. Prompts: "a bunch of people on skiing on a hill"; "A dog standing in front of a doorway."; "A man sitting at a table in front of bowls of spices."; "A person sitting on a motorcycle in the grass."; "A TV sitting on top of a wooden stand."; "Several bicycles sit parked nest to each other."; "A bunch of people posing with some bikes."; "Ornate archway inset with matching fireplace in room.".

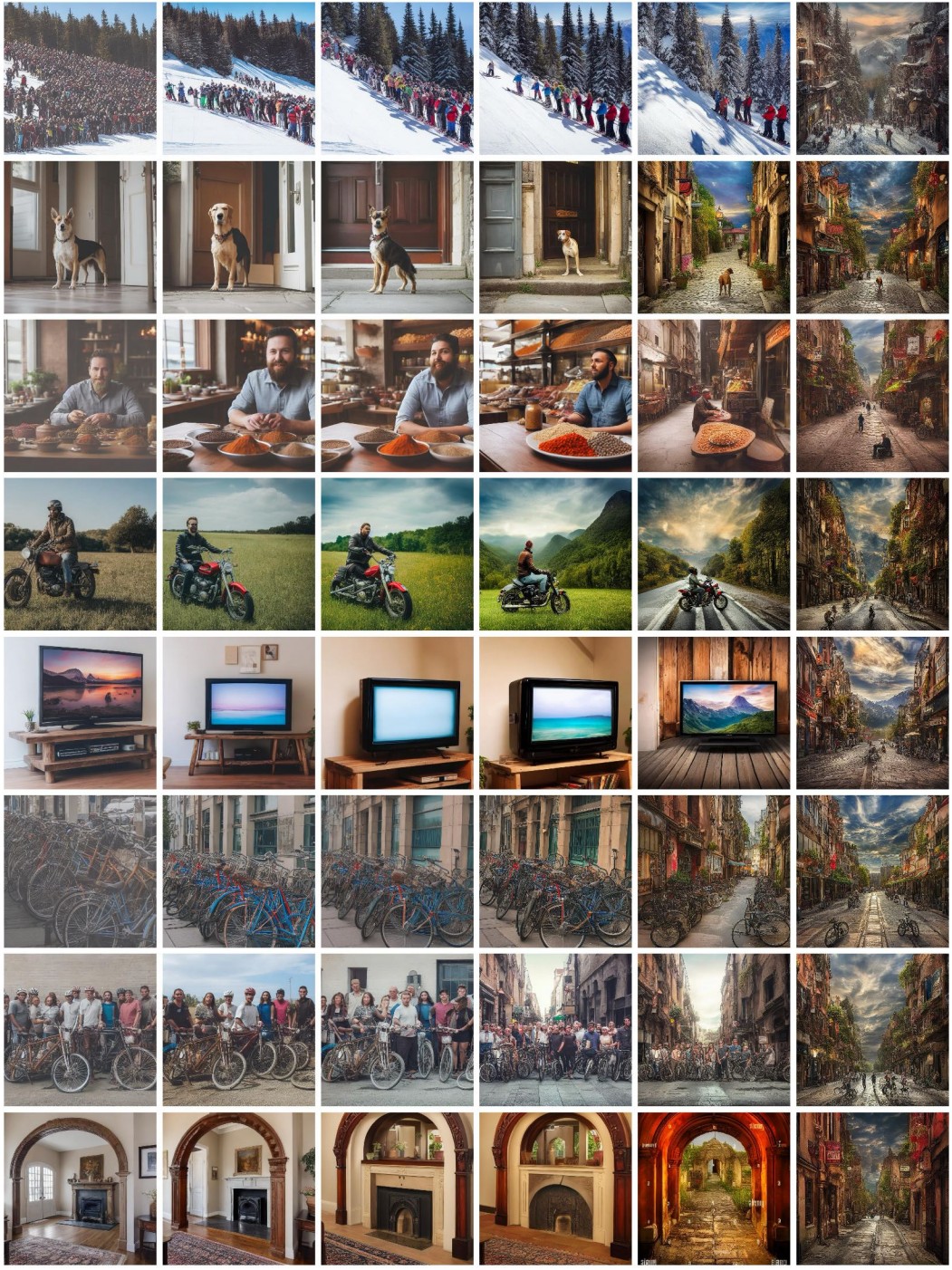

Figure 23: TDPO aligned with the aesthetic reward $f_1$ and PickScore reward $f_2$. The weights for $f_1$ and $f_2$ are $w_1$ and $w_2 = 1 - w_1$, respectively. From left to right, $w_1 = 0.0, 0.2, 0.4, 0.6, 0.8, 1.0$. Prompts: "a bunch of people on skiing on a hill"; "A dog standing in front of a doorway."; "A man sitting at a table in front of bowls of spices."; "A person sitting on a motorcycle in the grass."; "A TV sitting on top of a wooden stand."; "Several bicycles sit parked nest to each other."; "A bunch of people posing with some bikes."; "Ornate archway inset with matching fireplace in room.".

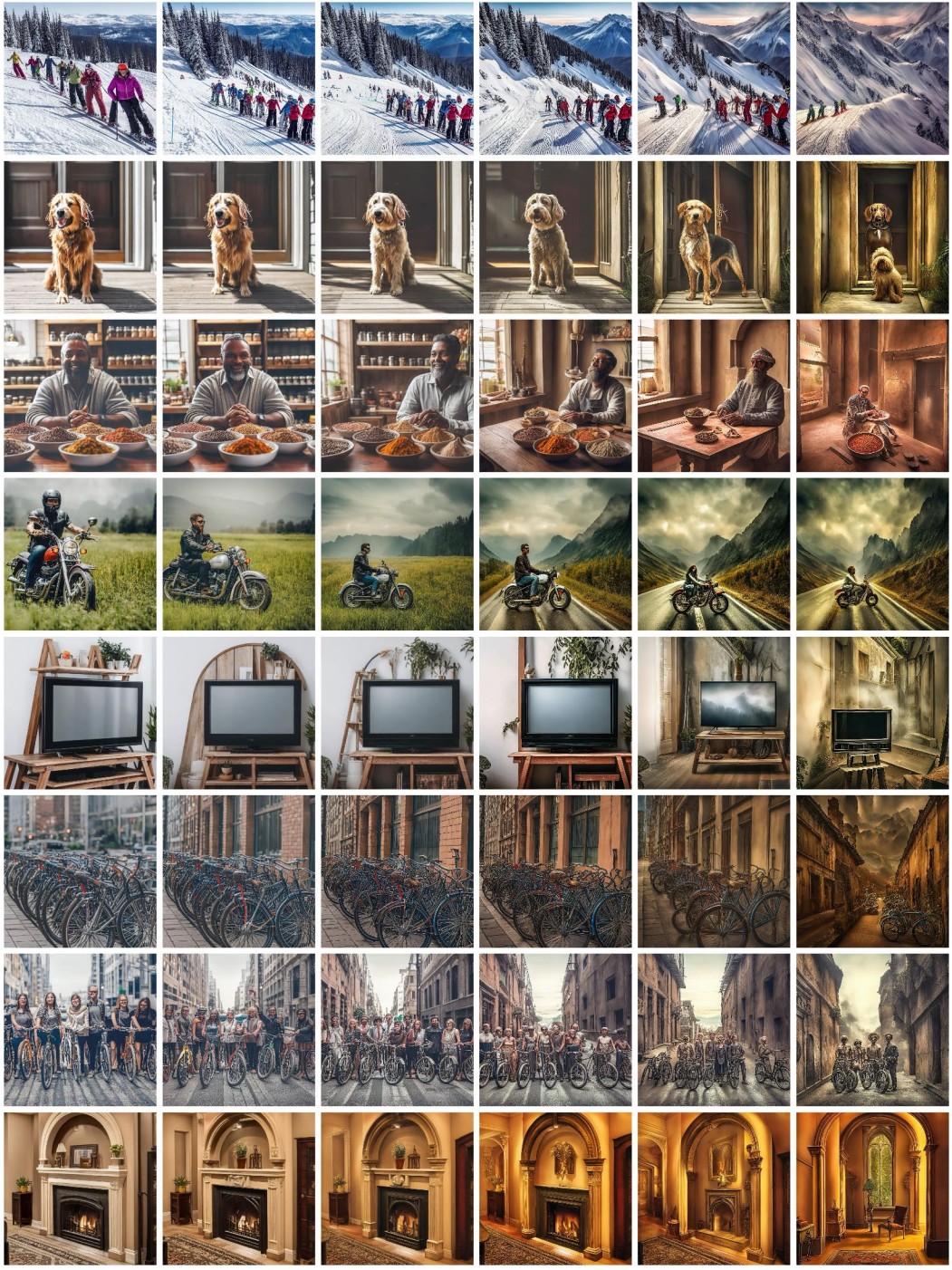

Figure 24: DDPO aligned with the aesthetic reward $f_1$ and PickScore reward $f_2$. The weights for $f_1$ and $f_2$ are $w_1$ and $w_2 = 1 - w_1$, respectively. From left to right, $w_1 = 0.0, 0.2, 0.4, 0.6, 0.8, 1.0$. Prompts: "a bunch of people on skiing on a hill"; "A dog standing in front of a doorway."; "A man sitting at a table in front of bowls of spices."; "A person sitting on a motorcycle in the grass."; "A TV sitting on top of a wooden stand."; "Several bicycles sit parked nest to each other."; "A bunch of people posing with some bikes."; "Ornate archway inset with matching fireplace in room.".

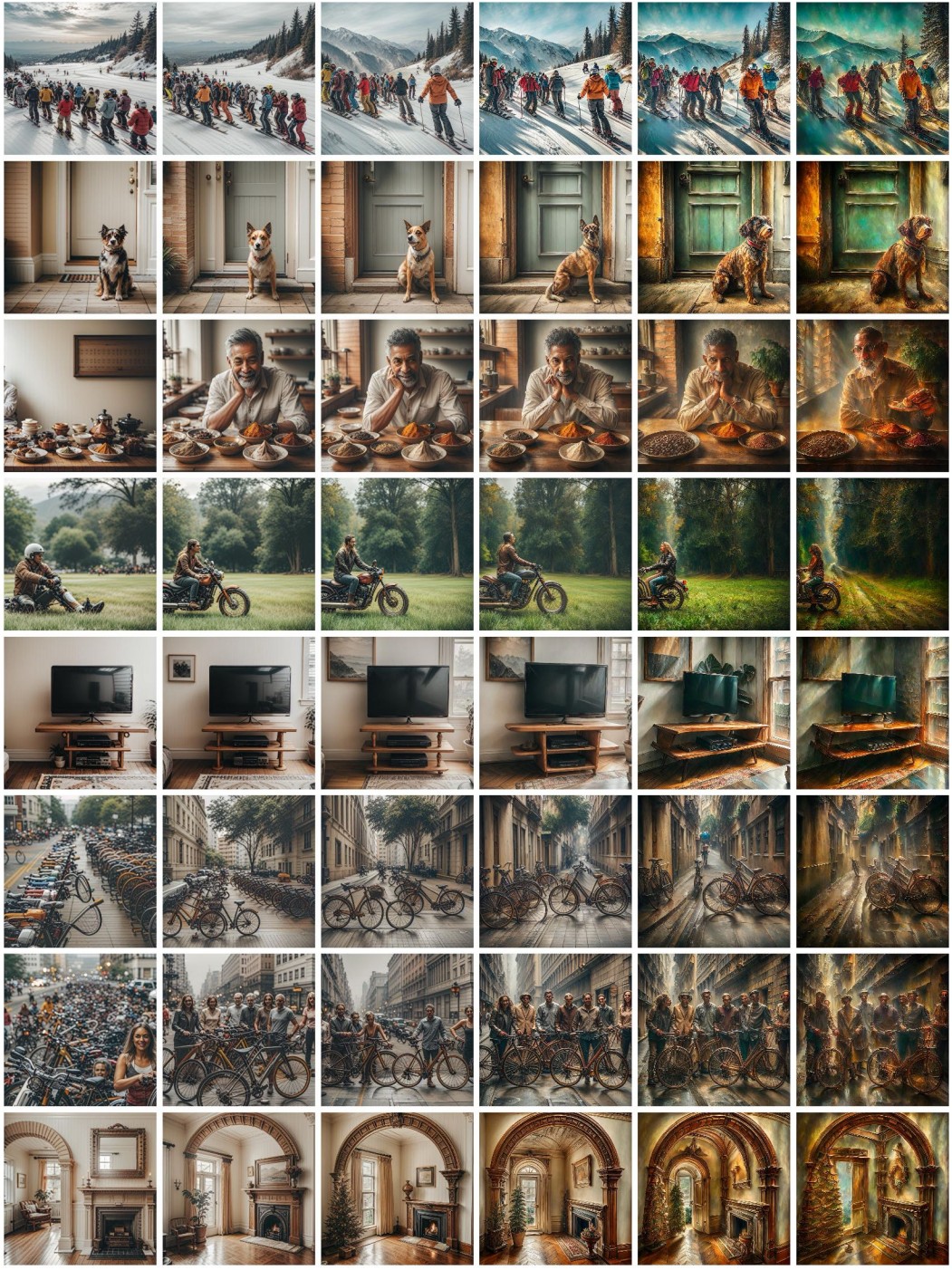

Figure 25: AlignProp aligned with the aesthetic reward $f_1$ and PickScore reward $f_2$. The weights for $f_1$ and $f_2$ are $w_1$ and $w_2 = 1 - w_1$, respectively. From left to right, $w_1 = 0.0, 0.2, 0.4, 0.6, 0.8, 1.0$. Prompts: "a bunch of people on skiing on a hill"; "A dog standing in front of a doorway."; "A man sitting at a table in front of bowls of spices."; "A person sitting on a motorcycle in the grass."; "A TV sitting on top of a wooden stand."; "Several bicycles sit parked nest to each other."; "A bunch of people posing with some bikes."; "Ornate archway inset with matching fireplace in room.".

