$$z^n = x_{t-1}^{m,n} - c_{t-1}^m, \text{ where } x_{t-1}^{m,n} \sim \mathcal{N}(c_{t-1}^m, \beta_t I),$$

where $\forall n = 1, 2, \ldots, N$.

Following Proposition 4, these samples $\{x_{t-1}^{m,n}\}_{n=1}^N$ can be also considered as sampling $\mathcal{N}(\mu_{t-1}, \beta_t I)$ when $\frac{\varepsilon}{\sqrt{\beta_t}}$ is sufficient small. So, we have

$$\bar{z}^n = x_{t-1}^{m,n} - \mu_{t-1} = z^n + c_{t-1}^m - \mu_{t-1}, \text{ where } x_{t-1}^{m,n} \sim \mathcal{N}(\mu_{t-1}, \beta_t