# OpenReview forum: "Sample Reward Soups: Query-efficient Multi-Reward Guidance for Text-to-Image Diffusion Models"
_ICLR.cc/2026/Conference — ICLR 2026 Poster_

### Official Review · Reviewer_5nQC · 2025-10-25

**Soundness:** 4
**Presentation:** 3
**Contribution:** 3
**Rating:** 6
**Confidence:** 3

**Summary:**

- This paper introduces SRSoup, a training-free and query-efficient diffusion-time alignment method for multiple rewards.
- The core idea is to avoid combinatorial blow-up by decomposing multi-objective optimization into $M$ simpler single-objective problems.
- The theory clarifies when the approximation is accurate: it holds when the objective’s curvature (second derivative) is small and the initialization is suitably chosen.
- In text-to-image experiments, SRSoup attains higher rewards under comparable query budgets.
- The empirical study also examines the roles of key hyperparameters.

Note: I used ChatGPT for minor language editing and phrasing assistance; all technical assessments are my own.

**Strengths:**

- The proposed multi-reward alignment method is training-free and query-efficient.
- Its query efficiency and accuracy are supported by both theory and experiments.
- The authors prove they can approximate the search gradient efficiently and accurately under stated assumptions, and they evaluate the finite-sample approximation error.
- The Pareto front is clearly visualized in experiments, and empirical Pareto near-optimality is demonstrated in Figures 3, 5, and 7.

Note: I used ChatGPT for minor language editing and phrasing assistance; all technical assessments are my own.

**Weaknesses:**

- As noted in lines 327–335, the main concern is that the assumption for Proposition 3 may not hold, leading to inaccurate approximations. The authors suggest switching from SRSoup to the standard method at the K-th step; however, the computational cost will then approach that of the standard “WeightedSum” method.
- For these reasons, I hope the computational cost as a function of K is clarified concretely in the main paper (I only found Table 4 in the appendix).
- The authors empirically validated density overlap using the Bhattacharyya coefficient (BC), whereas their theory uses Total Variation (TV). In the appendix and Proposition 4, they provide only an upper bound on TV. I think a lower bound on TV is also needed to justify the choice of K. The true TV might be small if BC is small.

Note: I used ChatGPT for minor language editing and phrasing assistance; all technical assessments are my own.

**Questions:**

- Could you extend your method to a per-subproblem optimization strategy with $M_2$ rewards per subproblem, where $M_2 < M$? I believe this could offer a favorable trade-off between quality and cost.
- For example, with 10 reward functions in total, you could approximately divide them into 5 subproblems, each containing 2 reward functions.
- Please note that I am NOT requiring any additional experiments for this.

Note: I used ChatGPT for minor language editing and phrasing assistance; all technical assessments are my own.

---

> ### Author Response · Authors · 2025-11-21
> **Rebuttal by Authors**
>
> **W1:** **switching at K makes cost approach weighted-sum**
>
> **Response:** Thanks for the comment. We clarify that switching from SRSoup to weighted-sum at the $K$-th denoising step does not cause the computational cost to approach that of the full weighted-sum method. The key reason is that the reward-query cost in the *early sampling stage**, where Proposition 4 holds and SRSoup is applied,* is reduced to the number of objectives. The choice of $K$ as 20 has been empirically validated by the Bhattacharyya coefficient (**Appendix Fig. 8**, see response to Reviewer `GEV9` W3 for details). With this setting, a significant amount of the computational savings are realized before the switch, reducing reward queries by up to 40\% compared to the weighted-sum strategy.
>
> Empirically, SRSoup still yields substantial efficiency gains: Table 4 shows significantly fewer reward queries than weighted-sum, and Fig. 1 demonstrates roughly a twofold improvement in query efficiency for comparable performance. Therefore, the computational benefits of SRSoup remain even though later steps follow the standard weighted-sum procedure.
>
> ---
>
> **W2: Clarify computational cost versus K in main paper**
>
> **Response:** We have included more results of time cost in **Fig. 1** in the main paper,  **Appendix Fig. 10** (HV versus $K$), **Table 5** for GPU memory, **Table 6** for inference time breakdown (can also see Table results in the response to Reviewer `GEV9` W1). In summary, the computational cost increase as $K$ decreases because reward queries increase.
>
> ---
>
> **W3:** **Need lower bound on TV; BC might underestimate**
>
> **Response:** Thanks for the comment. We clarify that TV can also be lower bounded by BC [1,2], namely, $1 - \mathrm{BC}(p,q) \le TV(p,q) \le \sqrt{2(1 - \mathrm{BC}(p,q))}$. Since TV has no closed-form expression for two general Gaussian distributions, BC is commonly used as an approximation. Moreover, BC is a similarity measure in the range $[0,1]$, making it convenient for comparison across different image-generation cases.
>
> Notably, we included the plot of BC values for $\mu_{t-1}$ and ${c}_{t-1}^m$ (**Appendix Fig.8**), justifying the choice of $K$ in our experiments.
>
> References
>
> [1] Bickel, P., Diggle, P., Fienberg, S., Gather, U., Olkin, I., & Zeger, S. (2009). *Principles and Theory for Data Mining and Machine Learning.* Springer.
>
> [2] [https://www.tcs.tifr.res.in/~prahladh/teaching/2011-12/comm/lectures/l12.pdf](https://www.tcs.tifr.res.in/~prahladh/teaching/2011-12/comm/lectures/l12.pdf).
>
> ---
>
>  **Q: Extending to per-subproblem optimization (M₂ < M)**
>
> **Response:** Thank you for this insightful suggestion. We agree that the proposed subproblem-based strategy could further reduce reward-query cost. One intuitive extension of SRSoup in this direction is to apply SRSoup separately to each subproblem to efficiently obtain local Pareto-optimal solutions. These solutions are then aggregated and evaluated with all reward models to approximate the global Pareto front. However, ensuring that the solutions obtained from the decomposed subproblems can recover the true global Pareto front remains a significant challenge and requires extensive investigation.

---

> > ### Comment · Reviewer_5nQC · 2025-11-26
> >
> > Thank you for the clear clarifications. The explanation about the cost efficiency and the BC–TV relationship resolved my concerns. I’m satisfied overall and will keep my positive score.

---

> > > ### Author Response · Authors · 2025-11-27
> > >
> > > Thank you for your time in reviewing our paper. We appreciate your positive assessment.

---

### Official Review · Reviewer_k88a · 2025-10-31

**Soundness:** 3
**Presentation:** 3
**Contribution:** 3
**Rating:** 6
**Confidence:** 4

**Summary:**

This paper proposes Sample Reward Soups (SRSoup), a training-free, query-efficient multi-reward alignment method for text-to-image diffusion models.
The core idea is to pre-compute per-reward gradients (“search gradients”) and to linearly interpolate them to approximate the gradients for arbitrary reward weightings.
By sharing reward evaluations across weights during the early diffusion steps—when the sample distributions are still overlapping—the method aims to achieve efficient Pareto-optimal sampling under multiple rewards.
Experiments on Stable Diffusion 1.5 and SDXL demonstrate similar or slightly better image quality compared to fine-tuning methods (e.g., DDPO, TDPO) while reducing reward queries by ~1.8–2.7×.

**Strengths:**

1. The paper tackles the important problem of reducing query computation cost in multi-objective preference optimization.

2. The number of queries can be explicitly calculated (as shown in Section C.3), and each parameter can be set according to the desired reduction target.

3. The proposed method is clearly formulated and easy to understand; unifying the Gaussian initialization to achieve a first-order gradient approximation is an interesting design.

**Weaknesses:**

1. The paper does not discuss the impact of varying K on performance.

2. There is no comparison of processing time with existing methods other than weighted-sum.

**Questions:**

1. How does the performance change when K is varied?

2. Would it be possible to compare the inference time with existing methods such as Rewarded Soups, AlignProp, TDPO, and DDPO?

---

> ### Author Response · Authors · 2025-11-21
> **Rebuttal by Authors**
>
> **W1&Q1: Effect of varying K (performance vs efficiency)**
>
> **Response:** Please note that the impact of varying $K$ on performance has already been analyzed in the original submission. As shown in Fig. 7, which reports the ablation study on the number of soup steps $K$, SRSoup achieves performance comparable to the weighted-sum baseline when $K \le 20$. As $K$ increases, the performance of SRSoup gradually degrades and becomes notably lower than that of weighted-sum sampling, confirming that large $K$ diminishes the effectiveness of exemplar-based gradients due to reduced trajectory overlap. We further include the plot of Hypervolume (HV) versus $K$ for quantatively demonstrating the performance of multi-reward alignment with different $K$ (**Appendix Fig. 10**).
>
> The efficiency implications of varying $K$ have also been examined (Table 4 in the original manuscript). Larger $K$ consistently yields substantial reductions in reward-query count and inference time. Together, these results highlight a clear trade-off: small $K$ provides stronger performance, whereas large $K$ improves query efficiency.
>
> ---
>
>  **W2&Q2: Compare inference time with Rewarded Soups, AlignProp, TDPO, DDPO.**
>
> **Response:** The alignment cost of finetuning-based methods (Rewarded Soups, AlignProp, TDPO, DDPO) comes entirely from their training stage, while their inference-time cost is essentially the same as the base model. In contrast, inference-time alignment methods like ours incur their alignment cost during inference, without requiring any training. Because these two classes of methods attribute cost to different stages, directly comparing inference-time latency would not be meaningful.

---

> > ### Author Response · Authors · 2025-11-27
> >
> > Dear Reviewer k88a,
> >
> > Thank you for your thoughtful and constructive comments, highlighting the value of our paper.
> >
> > We have carefully addressed your concerns regarding our submission. As the discussion period is ending soon, we would greatly appreciate your response to our rebuttal or any additional feedback. This would also allow us to address any remaining issues in a timely manner.
> >
> > We look forward to hearing from you.

---

### Official Review · Reviewer_itLM · 2025-10-31

**Soundness:** 3
**Presentation:** 3
**Contribution:** 3
**Rating:** 6
**Confidence:** 2

**Summary:**

This paper proposes a training-free multi-reward optimization framework that improves generated images of text-to-image diffusion models. In particular, the authors proposed a multi-reward alignment approach that optimize the denoising distribution at each sampling step using black-box optimization strategy. Additionally, they also proposed Sample Reward Soups, a mechanism that combines multiple reward objective that interpolates reward-guided search gradients. Experiment results show that the proposed inference-time framework achieves competitive performance compared to RL and other finetuning methods. They also demonstrated that the proposed Sample Reward Soups is more preferable than naive weighted sum of multiple rewards.

**Strengths:**

1. This paper is sufficiently novel in that it extends the idea of Reward Soap that interpolates multiple finetuned diffusion models to a training-free setup, making it more accessible while provided meaningful insight about the nature of this type of interpolation methods.
2. The author provided a detailed theoretical analysis of the characteristics of the proposed method, supported by toy experiments on mixture of Gaussian, which are informative.
3. The experiments are thorough. I especially appreciate the additional experiments and analysis in the supplementary material.

**Weaknesses:**

1. The author only showed experiments on SD-series U-Net, which is quite outdated at this time. I understand that many of the RL literature still focuses on SDv1.5 and SDXL due to computation constraints. However, as the proposed method is training free, it would be good to show results on state-of-the art rectified flow models based on diffusion transformers (DiT), such as Sana, SD3, Flux, etc. Such inclusion would make the paper more relevant.   It would also be interesting to see how the proposed method compared with the latest RL literature, such as Flow-GRPO. These latest works should also be discussed in the related works.

2. offline policy learning methods such as DPO are discussed but not included in the comparison. these model learns directly from human preference dataset which encompasses a diverse set of preference. It would be interesting to see however this method compare. While it is hard to apply soup to these models as they cannot be directly trained for different rewards, the author can still use other inference-time methods such as best-of-N sampling by matching the inference time compute.

3. for SDXL comparison in appendix, an RL baseline is missing, such as D3PO[2].


[1] Liu, Jie, et al. "Flow-grpo: Training flow matching models via online rl." arXiv preprint arXiv:2505.05470 (2025).
[2] Yang, Kai, et al. "Using human feedback to fine-tune diffusion models without any reward model." Proceedings of the IEEE/CVF Conference on Computer Vision and Pattern Recognition. 2024.

**Questions:**

I currently recommend this paper for weak acceptance.
I'm willing to increase my score based on author's responses. In particular, I hope the author can provide missing baselines and related literature.

---

> ### Author Response · Authors · 2025-11-21
> **Rebuttal by Authors**
>
> **W1: Apply SRSoup on flow models based on DiT; comparison with Flow-GRPO**
>
> **Response:** Thanks for the constructive suggestions.
>
> We have included the experiments of applying our SRSoup on the rectified flow models based on DiT, i.e., SD3. We also included the comparison with Flow-GRPO in the revised manuscript. Please see **Appendix C.2, C.3, Fig. 9(d), Fig. 13** for details. The results show the effectiveness of SRSoup on multi-reward alignment with the SD3 base, which further demonstrates the generality of our proposed method. SRSoup *outperforms Flow-GRPO* in terms of multi-reward alignment. Meanwhile, the visual results demonstrate that SRSoup achieves *better prompt alignment*.
>
> We have included the recent related works in the revised manuscript.
>
> ---
>
>  **W2: Missing comparison with offline policy learning method DPO**
>
> **Response:** As pointed out by the reviewer, DPO methods (e.g., DiffusionDPO, D3PO) are designed to guide T2I models using human preferences, which is not directly applicable to the setting of guiding T2I models using multi-reward models studied in this paper. To make a comparison, we follow D3PO, which adopts reward models for providing preference pairs. Specifically, the aesthetic reward model and compressibility reward model are used to provide preference pairs for fine-tuning SD1.5 with the DPO loss, respectively. Then similarly to DDPO (soup), we interpolate the two finetuned models for obtaining multi-reward guided samples. Please refer to **Appendix C.3 and Fig. 9(c)** for detailed experiments. The newly included results demonstrates that our SRSoup achieves better performance than D3PO (soup).
>
> ---
>
>  **W3: Missing D3PO baseline for SDXL**
>
> **Response:** Please note that D3PO is not implemented with the SDXL base. We included the comparison of SRSoup with D3PO on the SD1.5 base instead (see response to Weakness 2).

---

> > ### Author Response · Authors · 2025-11-27
> >
> > Dear Reviewer itLM,
> >
> > Thank you for your thoughtful and constructive comments, especially for highlighting the novelty of our proposed method and its theoretical and empirical support.
> >
> > We have provided the suggested baselines and related literature. As the discussion period is ending soon, we would greatly appreciate your response to our rebuttal or any additional feedback. This would also allow us to address any remaining issues in a timely manner.
> >
> > We look forward to hearing from you.

---

### Official Review · Reviewer_GEV9 · 2025-11-02

**Soundness:** 2
**Presentation:** 3
**Contribution:** 3
**Rating:** 6
**Confidence:** 3

**Summary:**

The paper Sample Reward Soups (SRSoup) addresses the challenge of multi-objective optimization in Text-to-Image diffusion models by proposing the first query-efficient, inference-time "soup" strategy. The core problem is that standard methods for balancing multiple black-box rewards (like aesthetic quality and compressibility) require an exponentially large number of expensive reward queries across the preference space. SRSoup solves this by independently calculating a reward-guided search gradient for each individual objective at every denoising step, and then linearly interpolating these gradients (not the model weights or rewards themselves). This allows the model to efficiently share and reuse sample rewards across different weighted objective combinations, drastically reducing the number of required queries and enabling superior or comparable performance to the Weighted Sum baseline.

**Strengths:**

1. Novelty of Inference-Time Gradient Interpolation: The paper introduces the first inference-time “soup” strategy (SRSoup) for diffusion process, effectively adapting the “Model Soups” concept not to model weights but to search gradients. This approach achieves Pareto-optimal sampling across a preference space without the risks of reward over-optimization often seen in fine-tuning methods.
2. Addresses a Scalability Bottleneck with High Efficiency: The paper addresses a limitation in aligning with multiple reward functions, which requires exponentially growing number of reward queries, which makes the process highly inefficient. The proposed SRSoup resolves this by significantly reducing the number of queries in the early denoising stages, establishing a more scalable solution for multi-objective T2I alignment.
3. Strong Empirical Results and Practical Impact: The method achieves significant query efficiency (e.g., up to 2.7x speedup) over the weighted-sum baseline, as demonstrated by the Hypervolume (HV) versus reward query plots.

**Weaknesses:**

1. Lack of Cost-Benefit Justification: While the paper shows a reduction in the number of reward queries, it fails to provide a comprehensive cost-benefit analysis for the entire inference pipeline. The reward query computation time, which is model-dependent, must be weighed against the overhead of running multiple parallel single-reward guidance steps required by SRSoup's gradient estimation.
2. Insufficient Analysis of Multi-Reward Trade-offs and Stability: The analysis, primarily relying on Hypervolume, does not rigorously demonstrate that SRSoup maintains a better balance or stability across the Pareto front compared to weighted-sum methods, especially with non-convex reward functions. This omission limits validation of the method's robustness against potential reward over-optimization.
3. Sensitivity and Justification of the Hybrid Strategy: The reliance on a hybrid strategy using SRSoup only for the first K steps lacks rigorous investigation into the determination and sensitivity of the boundary K.

**Questions:**

1. Please provide a detailed breakdown of the wall-clock time and GPU memory consumption across the full inference pipeline, explicitly weighing the reduced reward query time against the overhead of running M parallel guidance steps.
2. Given the risk of sub-optimal configurations in multi-objective optimization, how does SRSoup compare to simple weighted-sum methods in terms of output balance (i.e., avoiding cases where one reward is disproportionately maximized)?
3. Please provide a quantitative ablation study showing the performance (Hypervolume) and efficiency (Query Reduction) trade-offs across a range of K values (e.g., K=20, 40, 60).

---

> ### Author Response · Authors · 2025-11-21
> **Rebuttal by Authors**
>
> **W1&Q1: Cost–benefit justification, comparison on wall-clock time and GPU memory.**
>
> **Response:** We thank the reviewer for this constructive comment. We have added the time and GPU memory consumption results for our SRSoup and the weighted sum method. Another point worth highlighting is that reducing reward queries not only speeds up inference, but also cuts monetary costs when the reward models are commercial.
>
> Please refer to **Fig. 1** for the plot of HV versus inference time. With similar numbers of reward queries, the inference time is similar. We also provide inference-time (minute) breakdown required for one prompt under different numbers of reward queries as below (**Appendix Table 6**). "Total"refers to the total time for full pipeline inference. "Decode" refers to time spent decoding latent embeddings back into images for reward evaluation; *"*Reward*"* refers to time spent evaluating decoded images with reward models.
>
> | Method           | #Queries          | Aesthetic & Compressibility | Aesthetic & PickScore | Aesthetic & HPSv2   |
> | ---------------- | ----------------- | --------------------------- | --------------------- | ------------------- |
> |                  |                   | Total/Decode/Reward         | Total/Decode/Reward   | Total/Decode/Reward |
> | **SRSOUP** | 3000 ($K=50$)   | 9.2  / 8.8  / 0.2           | 9.7   / 9.0   / 0.3   | 9.7   / 9.0   / 0.3 |
> |                  | 5400 ($K=40$)  | 13.0 / 12.2 / 0.4           | 14.0  / 13.0 / 0.6    | 13.7  / 12.7 / 0.6  |
> |                  | 7800 ($K=30$)  | 17.1 / 15.9 / 0.6           | 17.4  / 16.2 / 0.8    | 17.6  / 16.2 / 0.9  |
> |                  | 10200 ($K=20$) | 20.4 / 19.2 / 0.8           | 21.6  / 20.0 / 1.1    | 21.6  / 20.0 / 1.2  |
> |                  | 12600 ($K=10$) | 24.3 / 23.1 / 0.9           | 25.1  / 23.3 / 1.4    | 25.4  / 23.4 / 1.6  |
> | **WS**     | 3000 ($N=6$)   | 8.2  / 7.5  / 0.3           | 9.3   / 8.4   / 0.5   | 9.4   / 8.5   / 0.5 |
> |                  | 5500 ($N=11$)  | 12.8 / 12.1 / 0.4           | 14.4  / 13.3 / 0.7    | 12.2  / 11.1 / 0.7  |
> |                  | 8000 ($N=16$)  | 17.7 / 16.8 / 0.6           | 18.4  / 17.0 / 1.0    | 15.7  / 14.3 / 1.0  |
> |                  | 10500 ($N=21$) | 21.7 / 20.4 / 0.9           | 23.0  / 21.4 / 1.2    | 21.1  / 19.2 / 1.4  |
> |                  | 13000 ($N=26$) | 24.8 / 23.4 / 1.0           | 27.1  / 25.3 / 1.4    | 25.8  / 23.7 / 1.7  |
>
> The peak GPU memory required for SRSoup and WeigthedSum (WS) when optimizing aesthetic \& compressiblity rewards on the animal prompts and when optimizing aesthetic \& PickScore rewards and aesthetic \& PickScore rewards on the HPD prompts (setting in **Fig. 1**) as below (**Appendix Table 5**). The GPU memory consumption of two methods is similar.
>
> |        | SRSoup | WS (N=26) | WS (N=21) | WS (N=16) | WS (N=11) | WS (N=6) |
> | ------ | ------ | --------- | --------- | --------- | --------- | -------- |
> | animal | 20.9GB | 20.9GB    | 20.9GB    | 20.9GB    | 20.9GB    | 18.4GB   |
> | HPD    | 24.6GB | 24.6GB    | 24.6GB    | 24.6GB    | 24.6GB    | 22.1GB   |
>
> ---
>
>  **W2&Q2: Multi-Reward Trade-offs and Stability; potential reward over-optimization.**
>
> **Response:** We appreciate the insightful comment.
>
> To clarify, we do not argue that SRSoup maintains a better balance or stability across the Pareto front compared to the weighted sum method in this work. Rather, SRSoup is proposed to reduce the reward-query burden by using sample reward soups to approximate the black-box gradient of the weighted-sum objectives (Proposition 3).
>
> We acknowledge that weighted-sum optimization can be sub-optimal, especially with non-convex reward functions. In our setting, the inherent stochasticity of diffusion sampling introduces noise-driven exploration, which helps escape poor local optima [1]. Regarding the risk that one reward becomes disproportionately maximized, this phenomenon is typically driven by differences in reward scales. To mitigate this, we applied reward normalization (Lines 943–944 in the manuscript), ensuring that no single reward dominates.
>
> Notably, the empirical results included in the original submission (**Fig. 3, Fig. 5, and Fig. 7**) show ***consistent improvement across all rewards***, demonstrating that SRSoup does not collapse optimization toward a single objective in practice.
>
> In addition, ***cross-reward evaluation*** included in the original submission (**Table 1, Table 2, Table 3**) demonstrates that our SRSoup does not suffer from reward over-optimization.
>
> [1] Liu, X., Tong, X., & Liu, Q. (2021). Profiling pareto front with multi-objective stein variational gradient descent. Advances in neural information processing systems, 34, 14721-14733.

---

> > ### Author Response · Authors · 2025-11-21
> > **Rebuttal by Authors (continued)**
> >
> > **W3&Q3: Justification of hybrid strategy & the choice of K**
> >
> > **Response:** Our adoption of hybrid strategy follows directly from **Proposition 4**, in which we proved that two sampling trajectories admit sufficient overlap for search-gradient reuse when their total variation distance (TV) remains small. TV has no closed form for two general gaussian distributions and Bhattacharyya coefficient (BC) is a common way to approximate it. We can use BC for a quantitative criterion for determining when exemplar-based gradients remain reliable.
> >
> > Specifically, we plot BC values for $\mu_{t-1}$ and $c_{t-1}^m$ (**Appendix Fig. 8**). At the beginning of the denoising process,
> > $\mathrm{BC}(c_{d,T-1}^m, \mu_{d,T-1}) = 1$ because $\mu_{T-1} = c_{T-1}^1 = \cdots = c_{T-1}^M$, due to identical initialization from $x_T \sim \mathcal{N}(0, I)$. At early stages ($t > 30$), over **50% of dimensions** satisfy $\mathrm{BC}(c_{d,t-1}^m, \mu_{d,t-1}) > 0.4$. At later stages ($t\to0$), this term approaches zero, making the approximation ineffective.
> >
> > For the quantitative sensitivity analysis of $K$, it has already been included in the original submission, i.e., **Fig. 7** (Pareto front) and **Table 4** (Query reduction). We included a new plot for HV versus $K$ (**Appendix Fig. 10**).
> > These ablations demonstrate the trade-off between performance and efficiency across different values of $K$ and substantiate the rationale chosen $K$ for our hybrid strategy.

---

> > > ### Author Response · Authors · 2025-11-27
> > >
> > > Dear Reviewer GEV9,
> > >
> > > Thank you for your thoughtful and constructive comments, particularly highlighting the novelty and efficacy of our proposed method.
> > >
> > > We have carefully addressed all of the concerns you raised regarding our submission. As the discussion period is ending soon, we would greatly appreciate your response to our rebuttal or any additional feedback, as this would allow us to address any remaining issues in a timely manner.
> > >
> > > We look forward to hearing from you.

---

### Author Response · Authors · 2025-11-21
**To all Reviewers**

We sincerely thank all reviewers for their insightful comments and constructive feedback. We appreciate the positive reception of our work, particularly the recognition of the novelty and query efficiency of our sample reward soup strategy. We have carefully merged all comments into our revised manuscript. The major updates are as follows:
* Include comparison on time and GPU for SRSoup and the weighted sum method (**Fig. 1, Appendix Table 5, Table 6**)
* Include empirical justification of the choice of soup steps $K$ (**Appendix Fig. 8**)
* Apply SRSoup to flow model based on DiT (SD3.5-M) (**Appendix C.2, Fig. 9(d), Fig. 13**)
* Compare with recent work Flow-GRPO, and D3PO for DPO objective (**Appendix C.3, Fig. 9(c)(d), Fig. 12, Fig. 13**); include the recent works in the literature review.

We are willing to discuss any further concerns or questions.

---

### Author Response · Authors · 2025-12-01
**To Area Chair: Summary of Review and Rebuttal**

**Dear Area Chair,**

We summarize the comments of reviewers and our responses for your reference.

All reviewers give positive assessments of our work (all with rating **6**). Particularly, they all acknowledge the technical contribution and strong empirical performance of our work.

* `GEV9`: "Addresses a Scalability Bottleneck with High Efficiency", "Strong Empirical Results and Practical Impact".
* `itLM`: "provided meaningful insight", "detailed theoretical analysis", "The experiments are thorough"
* `k88a`: "tackles the important problem of reducing query computation cost in multi-objective preference optimization"
* `5nQC`: "query efficiency and accuracy are supported by both theory and experiments"

Also, `GEV9` ("Novelty of Inference-Time Gradient Interpolation") and `itLM` ("sufficient novel"), and `k88a`  ("...achieve a first-order gradient approximation is an interesting design") recognize the novelty of our proposed method.

*`5nQC` acknowledged that our response well addressed their concerns, while the other reviewers have not yet replied to our responses.* Below, we summarize how we address the concerns raised by the reviewers.

* `GEV9`.
  * The GPU time and memory consumption of our proposed method: we include them as **Fig. 1, Appendix Table 5, Table 6**.
  * Multi-reward trade-offs and stability: we clarify that our method does not suffer from such issues with empirical support (results included in the original submission).
  * Justification of hybrid strategy and the choice of soup steps $K$: we clarify that our adoption of the hybrid strategy follows directly from Proposition 4 (included in the original submission). And we also provide empirical justification of the choice of $K$ (**Appendix Fig. 8**), as well as performance (HV) and efficiency (query reduction) across different $K$ (**Appendix Fig. 10, Table 6**).
* `itLM`.
  * Extend the proposed method to flow model: we apply our SRSoup to flow model based on DiT (SD3.5-M) (**Appendix C.2, Fig. 9(d), Fig. 13**).
  * More baselines and related literature: include the comparison with D3PO for DPO objective and recent work Flow-GRPO (**Appendix C.3, Fig. 9(c)(d), Fig. 12, Fig. 13**); include the missing recent work in the literature review.
* `k88a`.
  * The impact of varying $K$ on performance: we clarify it has been analyzed in the original submission and we also additionally include a plot of HV (multi-reward performance) versus soup step $K$ in **Appendix Fig. 10**.
  * Inference time comparison with fine-tuning-based method: we clarify that fine-tuning-based methods incur alignment cost only during training, so comparing their inference-time latency with our training-free method is not meaningful.
* `5nQC`.
  * The hybrid strategy with cost approaching the weighted-sum method: we elaborate on the reduction of computation cost with empirical support (results included in the original submission).
  * Computational cost versus $K$: we explain that the computational cost increase as $K$ decreases with newly included results in **Fig. 1, Table 5, Table 6**.
  * BC–TV relationship: we detail that BC is a commonly used approximation for TV.

We have carefully merged all comments into our revised manuscript (*colored the revision in red*) and summarized the major updates in `To All Reviewers`.

Thank you for your time in handling our submission.

Best Regards,
Authors

---

### Meta-Review · Area_Chair_GDoR · 2026-01-07

**Summary:**

All reviewers have given the paper positive ratings (all with 6) and consistently acknowledged its technical contributions, including improved efficiency (GEV9, k88a, 5nQC), rigorous theoretical analysis (itLM, 5nQC), and thorough experimental evaluation with strong empirical results (GEV9, itLM, 5nQC). The authors have carefully and convincingly addressed all concerns raised during the rebuttal process. In particular, for GEV9, k88a, and 5nQC, the authors added detailed runtime and memory analyses, provided clear justification for the choice of hyperparameter K, and discussed trade-offs among multiple reward signals. For itLM, they incorporated additional baseline comparisons (D3PO, DPO, and flow-GRPO) and extended the proposed method to flow-based models (SD3.5-M). Reviewer 5nQC have acknowledged that his concerns  were fully addressed in the comments. The authors also carefully revised the manuscript in response to all reviewer comments. Based on the strong technical quality, clarity of revisions, and positive consensus among reviewers, we recommend acceptance of the paper.

**Reviewer Concerns:**

Most of the concerns are addressed in the rebuttal by the authors.

**Reviewer Scores:**

All the reviewers give positive ratings for the papers. In addition, the authors do an excellent job in replying the concerns raised by the reviewers. I believe that all the reviewers would keep or upgrade their scores for the paper. (e.g., Reviewer 5nQC has shown his satisfaction for the rebuttal and keep his positive rating.)

---

### Decision · Program_Chairs · 2026-01-26

Accept (Poster)